# Sleep is bi-directionally modified by amyloid beta oligomers

**Güliz Gürel Özcan[1], Sumi Lim[1], Patricia LA Leighton[2,3], W Ted Allison[2,3], Jason Rihel[1]***

[1]Department of Cell and Developmental Biology, UCL, London, United Kingdom; [2]Centre for Prions & Protein Folding Disease, University of Alberta, Edmonton, Canada; [3]Department of Biological Sciences, University of Alberta, Edmonton, Canada

**Abstract** Disrupted sleep is a major feature of Alzheimer's disease (AD), often arising years before symptoms of cognitive decline. Prolonged wakefulness exacerbates the production of amyloid-beta (Aβ) species, a major driver of AD progression, suggesting that sleep loss further accelerates AD through a vicious cycle. However, the mechanisms by which Aβ affects sleep are unknown. We demonstrate in zebrafish that Aβ acutely and reversibly enhances or suppresses sleep as a function of oligomer length. Genetic disruptions revealed that short Aβ oligomers induce acute wakefulness through Adrenergic receptor b2 (Adrb2) and Progesterone membrane receptor component 1 (Pgrmc1), while longer Aβ forms induce sleep through a pharmacologically tractable Prion Protein (PrP) signaling cascade. Our data indicate that Aβ can trigger a bi-directional sleep/wake switch. Alterations to the brain's Aβ oligomeric milieu, such as during the progression of AD, may therefore disrupt sleep via changes in acute signaling events.

## Introduction

Accumulation of amyloid-beta (Aβ) in plaques, along with tau tangles, is one of the two pathological hallmarks of Alzheimer's disease (AD). Change in Aβ levels in the brain is one of the earliest known pathological events in AD and is detectable years before the development of Aβ plaques and decades before the clinical onset of AD (*Bateman et al., 2007*; *Jack et al., 2013*). Because of its importance in AD progression, Aβ has been mostly characterized as a functionless, pathological, intrinsically neurotoxic peptide (*Moir and Tanzi, 2019*). However, Aβ is an ancient neuropeptide conserved across vertebrates through at least 400 million years of evolution (*Moir and Tanzi, 2019*). Aβ's cleavage from amyloid precursor protein (APP) is tightly regulated by multiple enzymatic reactions (*O'Brien and Wong, 2011*), and its release from neurons is carefully controlled (*Kamenetz et al., 2003*). Aβ interacts with numerous surface receptors and can activate intrinsic cellular signalling cascades to alter neuronal and synaptic function (*Jarosz-Griffiths et al., 2016*). More recently, Aβ has been suggested to act as an antimicrobial peptide (*Soscia et al., 2010*), and the deposition of Aβ may be induced as an innate immune defence mechanism against microbial pathogens (*Kumar et al., 2016*). However, the various biological effects of Aβ in health or disease remain obscure.

One of the earliest symptoms of AD is the disruption of sleep, and AD patients have sleep-wake abnormalities, including insomnia at night and increased napping during the day (*Allen et al., 1987*; *Loewenstein et al., 1982*; *Moran et al., 2005*; *Prinz et al., 1982*). Multiple transgenic AD mouse models that overproduce Aβ also show disrupted sleep phenotypes (*Roh et al., 2012*; *Sterniczuk et al., 2010*; *Wang et al., 2002*), often in the absence of neuronal loss and preceding impairments of learning and memory (*Irizarry et al., 1997*). In non-pathological conditions, Aβ levels in the cerebrospinal fluid (CSF) are modulated by the sleep-wake cycle (*Kang et al., 2009*;

*For correspondence:
j.rihel@ucl.ac.uk

**Competing interests:** The authors declare that no competing interests exist.

*Xie et al., 2013*). Aβ generation and release are controlled by electrical and synaptic activity (*Cirrito et al., 2005*; *Kamenetz et al., 2003*), leading to increased extracellular Aβ levels during wakefulness and decreased levels during sleep (*Kang et al., 2009*; *Xie et al., 2013*). These observations have led to the proposal that sleep and Aβ dynamics create a vicious feed-forward cycle, wherein increases in wakefulness result in increased extracellular Aβ and aggregation, which then dysregulates sleep, further exacerbating pathogenic Aβ production (*Roh et al., 2012*). How increased Aβ burden leads to disruptions in sleep remains unknown, although AD-related cell death of critical sleep/wake regulatory neurons has been suggested as a possible mechanism (*Fronczek et al., 2012*; *Lim et al., 2014*; *Manaye et al., 2013*).

Given the relationship between Aβ and sleep, we hypothesized that Aβ may directly modulate sleep-regulatory pathways independently of neuronal cell death. To test this, we took advantage of the ability to directly deliver small molecules and Aβ peptides to the brain of larval zebrafish, which have conserved APP processing machinery and Aβ peptides (*Newman et al., 2014*) and share genetic, pharmacological, and neuronal sleep-regulatory mechanisms with mammals (*Barlow and Rihel, 2017*). We found that Aβ size-dependently and reversibly modulates behavior through two distinct genetic, pharmacologically tractable pathways that regulate sleep in opposing directions.

## Results

### Aβ dose-dependently modifies zebrafish sleep and wake behavior

Isolating the specific biological effects of Aβ has been experimentally difficult. One challenge is that Aβ is processed from a series of complex cleavage steps of a longer transmembrane protein, APP, which also produces other protein products with a variety of functions (*O'Brien and Wong, 2011*). This restricts the utility of genetic manipulations to tease out Aβ-specific roles from the other APP components. Another challenge is that Aβ forms, in vitro and in vivo, a variety of oligomeric species (e.g. dimers, longer oligomers, or large fibrils) with diverse structures, binding affinities, and signalling properties (*Benilova et al., 2012*; *Jarosz-Griffiths et al., 2016*). Teasing out the biological signalling capabilities of these diverse oligomeric species requires selective manipulation of Aβ oligomeric states, which is difficult in vitro and is currently nearly impossible endogenously in vivo.

To overcome some of these barriers, we developed an injection assay in which the amount and type of the Aβ oligomers can be controlled and then tested the acute signaling effects of Aβ on sleep and wake behavior. Our minimally invasive intra-cardiac injection assay in 5 days post fertilization (5 dpf) larval zebrafish avoids direct damage to brain tissue (*Figure 1A and B*). This technique rapidly (<1 hr, peaking within 2–3 hr) and reversibly delivers Aβ to the larval brain, as assessed by injection of fluorescently tagged Aβ42 and subsequent confocal brain imaging (*Figure 1B*, *Figure 1—figure supplement 1A,B*). To generate different Aβ oligomeric species, we modified previously established in vitro monomeric Aβ incubation protocols (see Extended Methods) that enrich for Aβ with different oligomeric sizes and opposing effects on rat neuronal excitability (*Kusumoto et al., 1998*; *Orbán et al., 2010*; *Whitcomb et al., 2015*). By incubating Aβ42 overnight at increasing temperatures, we generated Aβ oligomeric pools with significantly different lengths, as measured by transmission electron microscopy (TEM) (*Figure 1C* and *Figure 1—figure supplement 1C*). Aβ42 incubated overnight at 4°C consisted of fewer and shorter oligomers (Aβ$^{short}$, mean 45 ± 11 nm, median = 39 nm) than when incubated at 25°C (Aβ$^{long}$, mean 75 ± 10 nm, median = 61 nm) or at 37°C (Aβ$^{v-long}$, mean 121 ± 10 nm, median = 88 nm) (*Figure 1C*).

We then assessed how each Aβ preparation affected sleep and wake behavior in zebrafish relative to an Aβ42–1 'reverse' peptide control (Aβ$^{rev}$) using automated video-monitoring (*Prober et al., 2006*; *Rihel et al., 2010*). In initial experiments, we determined the appropriate Aβ injection dose by injecting 1 nL of a 1–1000 nM dose series for both Aβ$^{short}$ and Aβ$^{long}$ and assessing subsequent waking activity and sleep, which is defined in zebrafish larvae as a period of inactivity lasting longer than one minute, which are associated with an increased arousal threshold and other features of behavioral sleep (*Prober et al., 2006*). Unexpectedly, these oligomeric species had opposing behavioral effects (*Figure 1—figure supplement 1D–G*). Aβ$^{short}$ increased waking activity and decreased sleep relative to Aβ$^{rev}$ peptide, while Aβ$^{long}$ decreased waking and increased sleep (prep waking effect, p<0.001; prep sleep effect p<0.05, two-way ANOVA). These effects were generally consistent across doses, although some dose-responses elicited stronger differential effects than others

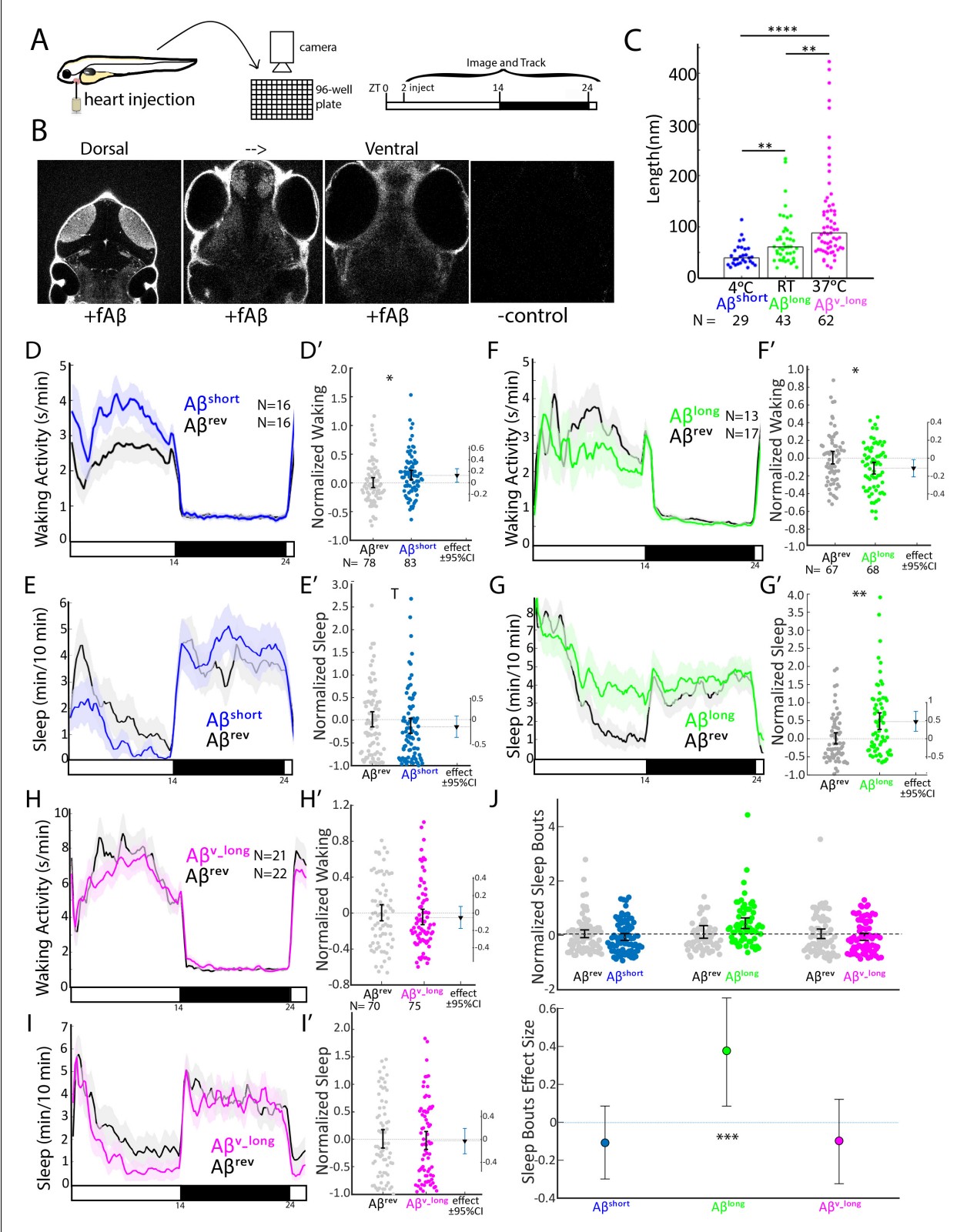

**Figure 1.** Aβ oligomers bi-directionally affect sleep and wake in zebrafish larvae. (**A**) Experimental schematic. Aβ was injected into the heart of 5 dpf larvae in the morning (ZT2 = zeitgeber time 2, that is 2hr after lights on). Behavior was then monitored in a square-welled 96-well plate for 24–48 hr on a 14 hr:10 hr light:dark cycle. (**B**) Heart-injected HiLyteTM Fluor 647-labeled Aβ42 (fAβ) penetrated the whole larval brain as visualized by confocal microscopy (optical sections, dorsal view) taken 2 hr after injection. Anterior is to the top. (**C**) Aβ prepared under increasing temperatures adopted

*Figure 1 continued on next page*

*Figure 1 continued*

longer oligomeric lengths, as measured by transmission electron microscopy. Each dot is a single oligomer (N = number measured), and the bars show the median. Data was taken from five randomly selected micrographs from two independent experiments. **p≤0.01, ****p≤1×10-7 Kruskal-Wallis, Tukey-Kramer post-hoc test. (D, E) Exemplar 24 hr traces post-injection comparing the effect of Aβ$^{short}$ (blue) on average waking activity (D) and sleep (E) versus Aβ$^{rev}$ controls (grey). Ribbons represent ±the standard error of the mean (SEM). Light and dark bars indicate the lights ON and lights OFF periods, respectively. N = the number of larvae in each condition. (D', E') The effect of Aβ$^{short}$ relative to Aβ$^{rev}$ on waking (D') and sleep (E') during the first day is shown, pooled from n = 5 independent experiments. Each dot represents a single larva normalized to the mean of the Aβ$^{rev}$ control, and error bars indicate ± SEM. The mean difference effect size and 95% confidence interval is plotted to the right. *p<0.05, $^{T}$p <0.1, one-way ANOVA. (F, G) Exemplar 24 hr traces post-injection comparing the effect of Aβ$^{long}$ (green) on average waking activity (F) and sleep (G) versus Aβ$^{rev}$ controls (grey). (F', G') The effect of Aβ$^{long}$ relative to Aβ$^{rev}$ on waking (F') and sleep (G') during the first day is shown, pooled from n = 4 independent experiments. *p<0.05, **p<0.01, one-way ANOVA. (H, I) Exemplar 24 hr traces post-injection comparing the effect of Aβ$^{v\_long}$ (magenta) on average waking activity (H) and sleep (I) versus Aβ$^{rev}$ peptide controls (grey). (H', I') The effect of Aβ$^{v\_long}$ relative to Aβ$^{rev}$ on waking (H') and sleep (I') during the first day is shown, pooled from n = 3 independent experiments. (J) The effect of different Aβ preparations on the number of sleep bouts relative to Aβ$^{rev}$ controls. The difference effect size and 95% confidence interval is plotted below. The asterisks indicate statistically significant different effects among the preps (***p<0.001, one-way ANOVA). See also *Figure 1—figure supplements 1–3*.

The online version of this article includes the following video and figure supplement(s) for figure 1:

**Figure supplement 1.** Aβ oligomers exert dose-dependent, short-term effects on zebrafish sleep.
**Figure supplement 2.** Aβ exposure does not increase neuronal cell death and does not alter survival into adulthood.
**Figure supplement 3.** Aβ-injected larvae recover after 24 hr and do not exhibit seizure-like or sickness behavior.
**Figure 1—video 1.** Aβ does not induce seizures.
https://elifesciences.org/articles/53995#fig1video1

---

(*Figure 1—figure supplement 1D,E*), with the maximal difference between the Aβ$^{short}$ and Aβ$^{long}$ preparations at 10 nM (p≤0.01 doseXprep interaction, two-way ANOVA). We estimate this dose yields a final concentration that falls within the lower range of physiological concentrations reported for Aβ42 in human CSF of 100 pM-5nM (*Bateman et al., 2007*). For example, assuming that all injected Aβ goes into the brain, the highest possible concentration would be 1500 pg/ml or 300 pM (45 ng/ml x 1 nl in 30.4 nl brain = 1.5 ng/ml = 1500 pg/ml). At the lower end, assuming equal distribution of Aβ over the whole body yields a final concentration estimate of 150 pg/ml or 30 pM (45 ng/ml x1 nl in 300 nl of body = 150 pg/ml). We therefore continued with 10 nM injections for all subsequent experiments, as it combines the maximal differentially observed behavioral effects between Aβ$^{short}$ and Aβ$^{long}$ with physiologically reasonable concentrations.

## Aβ affects sleep and wake in opposing directions as a function of oligomer size and independently of neural death

To explore the effect of Aβ oligomeric size on sleep, we then systematically tested the behavior effects of each Aβ species relative to a Aβ$^{rev}$ control for n = 3–5 independent experiments each. As in the dose response experiments, Aβ affected sleep and wake in opposing directions depending on its oligomeric state (*Figure 1D–I'*). In the day following injection, Aβ$^{short}$ significantly increased waking activity by +12.8% and reduced total sleep relative to Aβ$^{rev}$ by 15.5% (*Figure 1D–E'*). The magnitude of the sleep effect is likely partially masked by a flooring effect due to generally reduced sleep during the day; we therefore favor reporting effect sizes and confidence intervals as recommended (*Amrhein et al., 2019*; *Ho et al., 2019*). Indeed, if we were to combine all the additional control Aβ$^{short}$ experiments subsequently reported in this manuscript (n = 160 Aβ$^{rev}$ n = 164 Aβ$^{short}$, see Figure 3G and H), the effect size remains robust at −15.9% and the result is statistically significant (p<0.05, one-way ANOVA). These effects were reversible, as there were no significant differences in sleep (*Figure 1E*, black bar) or waking activity (*Figure 1D*, black bar) between Aβ$^{short}$ and reverse peptide in the night following injection, and the behavior of Aβ$^{short}$-injected larvae returned to baseline levels in the subsequent day (*Figure 1—figure supplement 3A*).

In contrast, while injection of longer Aβ fibers (Aβ$^{v\_long}$) had no effect on behavior, (*Figure 1H–I'*), injection of the intermediate Aβ$^{long}$ oligomers significantly increased sleep during the post-injection day by +47.2% and reduced waking activity by 11.3% (*Figure 1F–G'*). The increased sleep induced by Aβ$^{long}$ was due to a significant increase in the average number of sleep bouts but not an increase in sleep bout length (*Figure 1J*), indicating higher sleep initiation is responsible for the change in sleep rather than an increased sleep consolidation. This increased sleep effect by Aβ$^{long}$ was not

observed in the night following injection (*Figure 1F and G*, black bar), and behavior returned to baseline by the following morning (*Figure 1—figure supplement 3B*).

This data is consistent with Aβ^short increasing wakefulness and Aβ^long decreasing wakefulness and increasing sleep. Additional control experiments ruled out experimental artefacts, as larvae undergoing no treatment, anesthesia only, mock injection, or PBS only injections had indistinguishable effects on sleep/wake (*Figure 1—figure supplement 1H–J*). Next, we recalculated the behavioral analysis only for the evening period before lights off, when vehicle-injected larvae were statistically indistinguishable from larvae that had been acclimated to the tracking rig for 24 hr (*Figure 1—figure supplement 1J*). Except for an even more severe flooring effect in the Aβ^short injection experiments, the results from evening-only analysis were indistinguishable from calculations across the whole day (*Figure 1—figure supplement 1K*). We therefore used full day analysis for all subsequent experiments.

We next considered if the dual-effects of Aβ on sleep and wake are due to either neuronal damage or generalized toxic effects, such as the induction of seizure, paralysis, or sickness behavior.

First, injection with either long or short forms of Aβ had no effect on apoptosis, as detected by staining for activation of Caspase-3 (*Figure 1—figure supplement 2A–C*). In addition, Aβ injected animals raised to adulthood showed no major differences in their general health or in their survival rates (*Figure 1—figure supplement 2D*). Moreover, injected animals recovered fully in the long term, returning to baseline sleep and activity levels within 24 hr (*Figure 1—figure supplement 3A, B*). Second, both Aβ^short and Aβ^long injected larvae responded normally to salient stimuli such as a light:dark pulse, demonstrating that these larvae were not paralyzed, in a coma, or undergoing sickness behavior (*Figure 1—figure supplement 3C*). Finally, we considered if the changes in motility in Aβ-injected larvae were seizure-like behaviors. Wild type (WT) zebrafish larvae display 'burst-and-glide' movements characterized by single short forward or turn movement followed by a short pause (*Figure 1—figure supplement 3D* and *Figure 1—video 1*). In contrast, epileptogenic drugs like the GABA-receptor antagonist PTZ induce electrophysiological and behavioral seizures (*Baraban et al., 2005*), which are observed as dramatic rearrangements in zebrafish bout structure (*Figure 1—figure supplement 3D*). The bout structure of Aβ^rev, Aβ^short, and Aβ^long injected fish was highly similar to WT behavior (*Figure 1—figure supplement 3D,E* and *Figure 1—video 1*), and the high-frequency bouts (HFB) indicative of seizures (*Reichert et al., 2019*) were only found in PTZ exposed fish but not Aβ injected larvae (*Figure 1—figure supplement 3D,E*). Together these experiments indicate that exposure to Aβ modulates normal sleep/wake behavior without inducing toxic states.

We conclude that the changes in behavior after Aβ exposure are due to acute signalling events and therefore sought to identify the neuronal and molecular substrates through which Aβ signals to modulate sleep/wake behavior.

## Aβ^short and Aβ^long induce opposing changes in neuronal activity and differentially engage sleep-promoting neurons

If Aβ oligomers alter behavior through acute signaling in the brain, the differential effects of Aβ^short and Aβ^long should be reflected at the level of neuronal activity. In situ hybridization (ISH) for expression of the immediate early gene, *c-fos*, identified several discrete areas of the larval brain that are upregulated after injection of Aβ^short relative to Aβ^rev, including the posterior hypothalamus and the dorsal and ventral telencephalon (*Figure 2A*), areas that are also upregulated in mutants with extended wakefulness (*Ashlin et al., 2018*). Comparing the Aβ^short induced *c-fos* patterns to WT brains collected at zeitgeber time 1 (ZT1, ZT0 = lights ON), when larvae are maximally awake, reveals at least nine populations of *c-fos*-positive neurons in both Aβ^short and waking brains (*Figure 2A,C*). In contrast, *c-fos* expression following Aβ^long injections was globally dampened relative to Aβ^rev (*Figure 2B*) in a manner consistent with the low expression of *c-fos* in WT brains collected at ZT19, when larvae are maximally asleep (*Figure 2C*).

Immediate early gene expression is an imperfect readout of changes in neuronal activity and brain state, as baseline *c-fos* is expressed in low amounts in zebrafish and has a relatively slow time course of 15–30 min for transcription of mRNA (*Baraban et al., 2005*). We therefore also quantified changes in the more rapid (<5 min) neuronal activity marker, phosphorylated ERK (p-ERK), using the larval zebrafish MAP-Mapping technique (*Randlett et al., 2015*). This method identifies the relative quantitative changes in brain region-specific levels of p-ERK relative to total ERK between Aβ injections and reverse peptide control conditions. Consistent with *c-fos* induction, Aβ^short upregulated P-ERK in the ventral telencephalon and posterior hypothalamus (*Figure 2D and D'*, *Figure 2—*

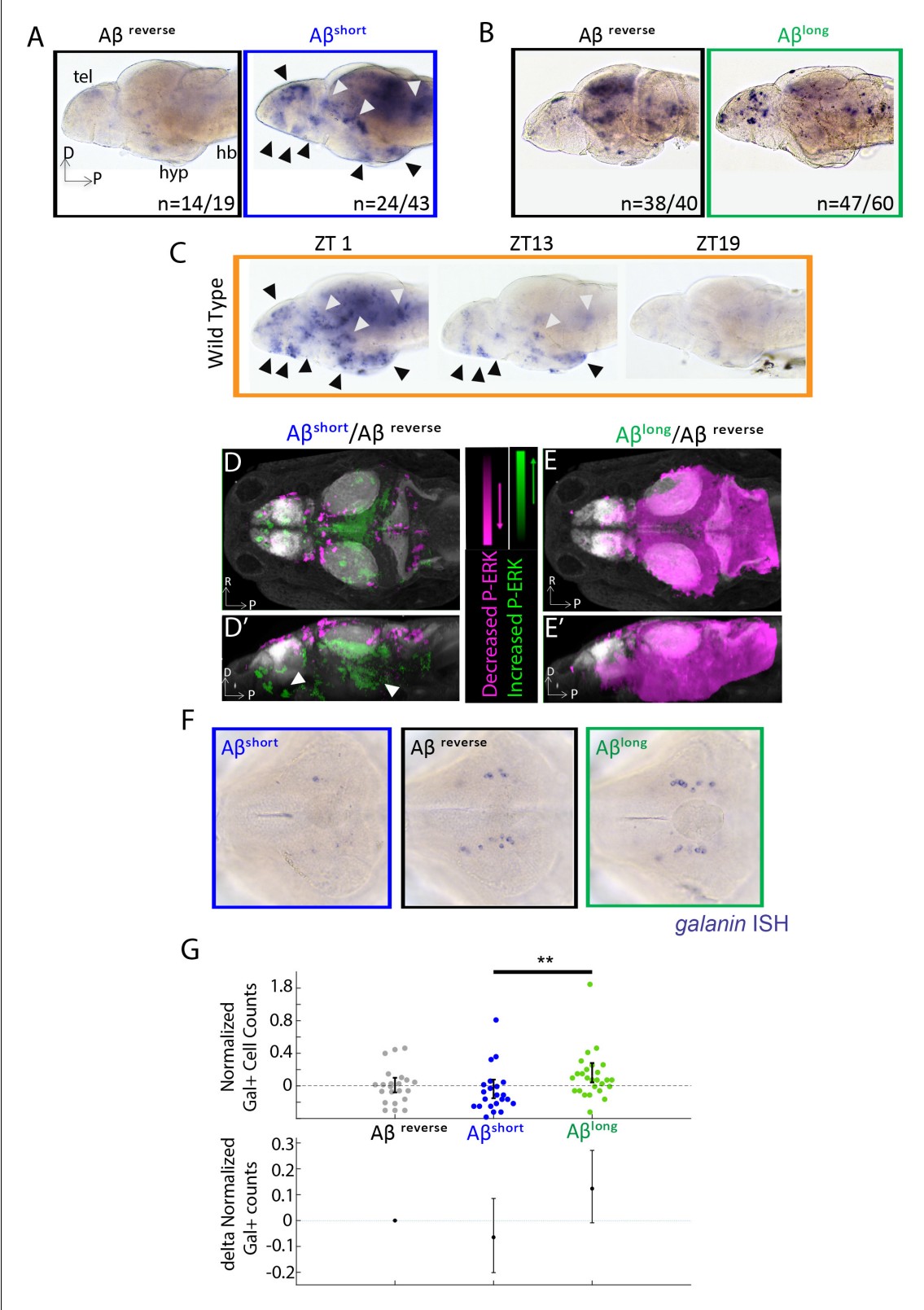

**Figure 2.** Aβ oligomers differentially alter neuronal activity in the larval zebrafish brain. (**A**) As detected by ISH, the immediate early gene *c-fos* is upregulated in many larval brain areas following Aβ<sup>short</sup> injection, including the dorsal and ventral telencephalon (tel) and the posterior hypothalamus (black arrowheads), relative to Aβ<sup>rev</sup> control injections. Other upregulated areas in the midbrain and hindbrain are indicated (white arrowheads). hyp-hypothalamus; hb- hindbrain. D = dorsal, p=Posterior, R = Right. n = blind counts of brains with the shown expression pattern/total brains. 24/43

*Figure 2 continued on next page*

Figure 2 continued

stringently counts only brains with the major areas upregulated. (B) Compared to Aβ$^{rev}$ injections, Aβ$^{long}$ oligomers induce less *c-fos* expression. The Aβ$^{rev}$ and Aβ$^{long}$ treated brains were stained longer than in (A) to ensure detection of weaker *c-fos* expression. n = blind counts of number of brains with the shown expression/total brains. (C) *c-fos* is upregulated in many larval brain areas at 10 am (ZT1) awake fish, including the dorsal and ventral telencephalon and the posterior hypothalamus (black arrowheads), and other discrete regions of the mid and hindbrain (white arrowheads). *c-fos* expression is downregulated in later timepoints (ZT13) and is very low in ZT19 brains, when larvae are predominantly asleep. N = 10 fish/timepoint. (D, D′) Brain expression of the neuronal activity correlate pERK/tERK comparing Aβ$^{short}$ (n = 6) to Aβ$^{rev}$ (n = 5) injected larvae identified areas upregulated (green) and downregulated (magenta) by Aβ$^{short}$. Data are shown as a thresholded maximum projection overlaid on the Z-Brain Atlas tERK reference (gray). White arrowheads indicate regions in the ventral telencephalon and posterior hypothalamus that are upregulated similar to *c-fos* in (A). Dorsal view in (D), lateral view in (D′). (E, E′) pERK/tERK expression after Aβ$^{long}$ injections (n = 7) shows widespread downregulation of neuronal activity (magenta) compared to Aβ$^{rev}$ controls (n = 7), consistent with *c-fos* data in (B). Dorsal view in (E), lateral view in (E′). (F) As detected by ISH, the number and intensity of hypothalamic *galanin*-positive neurons are downregulated following Aβ$^{short}$ injection and upregulated following Aβ$^{long}$ injection, relative to Aβ$^{rev}$ control injections. Representative images from N = 22–24 per condition. (G) Normalized, blinded counts of hypothalamic *galanin*-positive cell numbers 4–6 hr after Aβ$^{short}$ and Aβ$^{long}$ injections, relative to Aβ$^{rev}$. Error bars indicate ± SEM. The mean difference effect size and 95% confidence interval is plotted at the bottom. \*\*p<0.01, one-way ANOVA. See also *Figure 2—source datas 1* and *2*.

The online version of this article includes the following source data for figure 2:

**Source data 1.** MAP-Mapping of brain areas that are significantly up- and down-regulated in P-ERK levels in response to Aβ$^{short}$.

**Source data 2.** MAP-Mapping of brain areas that are significantly up- and down-regulated in P-ERK levels in response to Aβ$^{long}$.

*source data 1*), while Aβ$^{long}$ resulted in a widespread reduction in p-ERK levels throughout most of the brain (*Figure 2E and E′*, *Figure 2—source data 2*). These brain activity states are consistent with the induction of wakefulness by Aβ$^{short}$ and sleep by Aβ$^{long}$.

Finally, if the behavioral states induced by Aβ are bona fide sleep/wake states, we reasoned that known zebrafish sleep/wake regulatory neurons should be engaged. Galanin-expressing neurons of the preoptic area and hypothalamus are active and upregulate *galanin* transcription during zebrafish sleep (*Reichert et al., 2019*). Similarly, ISH for galanin 4–6 hr post-injection of Aβ oligomers revealed that wake-promoting Aβ$^{short}$ slightly decreased (−6%, blinded counts), while sleep-promoting Aβ$^{long}$ slightly increased (+12%, blinded counts), the number of *galanin*-positive cells in the hypothalamus compared to Aβ$^{rev}$ injected larvae (*Figure 2F–G*). The differential effects on *galanin* neurons are consistent with that the induction of wakefulness by Aβ$^{short}$ and sleep by Aβ$^{long}$.

## Aβ binding targets are required for behavioral responses to Aβ

Many candidate Aβ binding partners have been implicated in mediating the signalling effects of Aβ on synapses, with some targets showing preferences for Aβ dimers, such as Adrenergic Receptor β2 (ADRB2) (*Wang et al., 2010*), or low molecular weight (50–75 kDa) species, such as the Progesterone Membrane Receptor Component 1 (PGRMC1) (*Izzo et al., 2014b*), while other targets preferentially bind to longer oligomers/protofibrils, such as the Prion Protein (PrP) (*Laurén et al., 2009*; *Nicoll et al., 2013*). We therefore used Crispr/Cas9 to make genetic lesions in several zebrafish candidate Aβ receptors, choosing examples with reported affinities for various sized Aβ oligomers (*Figure 3—figure supplement 1* and *Figure 4—figure supplement 1*). We isolated a *pgrmc1* allele with a 16 bp deletion that leads to a frameshift and early stop codon that truncates the protein before a conserved Cytochrome b5-like Heme/Steroid binding domain (*Figure 3—figure supplement 1A–D*). We also isolated an *adrb2a* allele with an 8 bp deletion that leads to a severely truncated protein lacking all transmembrane domains (*Figure 3—figure supplement 1E–G*). We also obtained a *prp1$^{-/-}$;prp2$^{-/-}$* double mutant (*Fleisch et al., 2013*; *Leighton et al., 2018*) that lacks both zebrafish Prp proteins with conserved Aβ binding sites (*Figure 4—figure supplement 1*). The third zebrafish *prion* gene product, Prp3, does not have the conserved Aβ binding domains present in Prp1 and Prp2 (*Figure 4—figure supplement 1*). Except for a mild increase in daytime sleep in *adrb2a$^{-/-}$* mutants (*Figure 3—figure supplement 2D′–F′*), none of these mutants exhibited changes in baseline sleep and wake on a 14 hr:10 hr light:dark cycle as compared to wild type and heterozygous siblings (*Figure 3—figure supplement 2A–F′*; *Figure 4—figure supplement 2A–F′*). Under baseline conditions, we also detected no significant differences in day or night sleep and waking activity in *prp1$^{-/-}$;prp2$^{-/-}$* double mutants compared to *prp$^{+/+}$* siblings generated from either *prp1$^{+/-}$; prp2$^{+/-}$* or *prp1$^{+/-}$;prp2$^{-/-}$* in-crosses (*Figure 4—figure supplement 2A–F′*). The mild baseline phenotypes

allowed us to test the effect of Aβ oligomers in these mutants without complex behavioral confounds.

We first tested the effects of Aβ$^{short}$ injection on mutant behavior. Unlike the wild type controls, neither the *adrb2a*$^{-/-}$ nor the *pgrmc1*$^{-/-}$ mutants increased waking activity (*Figure 3A–C and E*) or suppressed sleep as observed in wild-type controls (*Figure 3A'–C' and G*). Injection of Aβ$^{short}$ into *adrb2a*$^{-/-}$ animals even significantly increased sleep (+83.7%) instead of reducing it as in wild-type larvae (*Figure 3B' and G*). In contrast, Aβ$^{short}$ injected into mutants that lack both zebrafish Prp orthologs (*prp1*$^{-/-}$; *prp2*$^{-/-}$) elicited slightly stronger increases in waking activity and significantly large (−45%) reductions in sleep (*Figure 3D,D', F and H*). Thus, the wake-promoting activity of Aβ$^{short}$ requires intact Adrb2a and Pgrmc1 but not functional Prp1 and Prp2.

Because the size of oligomeric species in our Aβ$^{long}$ preparation (20–400 nm) falls into the size range that exhibits high-affinity binding to mammalian Prion Protein (PrP) (20–200 nm) and thereby acts to modulate synapses (*Gimbel et al., 2010*; *Laurén et al., 2009*; *Nicoll et al., 2013*; *Um et al., 2012*), we tested whether PrP is instead required for Aβ$^{long}$-induced sleep. After injection of Aβ$^{long}$, *prp1*$^{-/-}$;*prp2*$^{-/-}$ null mutants failed to increase sleep compared to wild-type controls (*Figure 4A–D*). The modest reduction of wakefulness induced by Aβ$^{long}$ was even reversed in *prp1*$^{-/-}$;*prp2*$^{-/-}$ mutants, with Aβ$^{long}$ instead significantly increasing wakefulness (*Figure 4C*). Thus, while Prps are not required for the wake-inducing effects of Aβ$^{short}$, functional Prp1 and Prp2 are essential for sleep induced by Aβ$^{long}$. Moreover, since *prp* double mutants have exacerbated wakefulness in response to Aβ$^{short}$ injections, the sleep-inducing Prp pathway is likely co-activated along with the wake-promoting pathway by Aβ to partially dampen the behavioral response of wild-type larvae (*Figure 3D and H*).

## Mutants lacking Aβ targets have altered brain activity in response to Aβ consistent with behavioral effects

If Aβ$^{short}$ interacts with Adrb2a and Pgrmc1 to drive changes in wakefulness, the increased neuronal activity we observed in wild-type larvae after Aβ$^{short}$ injections (*Figure 2A*) should also be abolished in the *adrb2a*$^{-/-}$ and *pgrmc1*$^{-/-}$ mutant backgrounds. Consistently, lack of either Adrb2a or Pgrmc1 abolished the neuronal activity-inducing effect of Aβ$^{short}$ (*Figure 5A*), as detected by in situ hybridization for *c-fos*. In particular, the neuronal activity observed in the posterior hypothalamus and the dorsal and ventral telencephalon after Aβ$^{short}$ into WT controls was not detected after injection into either *adrb2a*$^{-/-}$ or *pgrmc1*$^{-/-}$ mutants (*Figure 5A*). This result is consistent with Aβ$^{short}$ failing to induce wakefulness in these mutants. Similarly, although Aβ$^{long}$ dampens neuronal activity when injected into wild-type larvae, Aβ$^{long}$ injections into the *prp1*$^{-/-}$; *prp2*$^{-/-}$ double mutants elicited no reduction in *c-fos* expression (*Figure 5B*). Instead, *c-fos* expression in the telencephalon and hypothalamus was upregulated relative to reverse injected controls (*Figure 5B*), consistent with the increased behavioral wakefulness observed in the *prp* mutants (*Figure 4C*). Together, these data are consistent with Aβ$^{short}$ acutely upregulating neuronal activity and behavioral wakefulness through interactions with Adrb2a and Pgrmc1, while Aβ$^{long}$ interactions with Prp drive increased sleep and a global reduction in neuronal activity.

## Pharmacological blockade of the Prp-mGluR5-Fyn kinase signaling cascade prevents sleep induction by Aβ$^{long}$

One of the advantages of using the zebrafish model system is the ability to perturb Aβ signalling cascades with small molecule inhibitors added directly to the water (*Kokel et al., 2010*; *Rihel et al., 2010*). To further dissect the Aβ$^{long}$-PrP sleep-inducing pathway, we focused on disrupting the putative signalling cascade downstream of Aβ-Prp interactions that lead to synaptic changes in neuronal culture (*Um et al., 2012*; *Figure 6A*). Consistent with a role for direct Aβ$^{long}$-Prp interactions in sleep, soaking the larvae in Chicago Sky Blue 6B, a small molecule reported to disrupt Aβ-PrP interactions (*Risse et al., 2015*), significantly abolished the sleep-inducing effect of Aβ$^{long}$ (*Figure 6B and C*, S7A and *Figure 6—figure supplement 1A,B*). Similarly, pharmacological inhibition of either of the putative Aβ-Prp downstream signalling components Metabotropic Glutamate Receptor 5 (mGluR5) or Fyn kinase (*Um et al., 2013*; *Um et al., 2012*) significantly blocked the sleep-inducing properties of Aβ$^{long}$ (*Figure 6D and E*, *Figure 6—figure supplement 1C,D*). Both the mGluR5 inhibitor MPEP and the Src-kinase inhibitor saracatinib even resulted in significant sleep reductions after

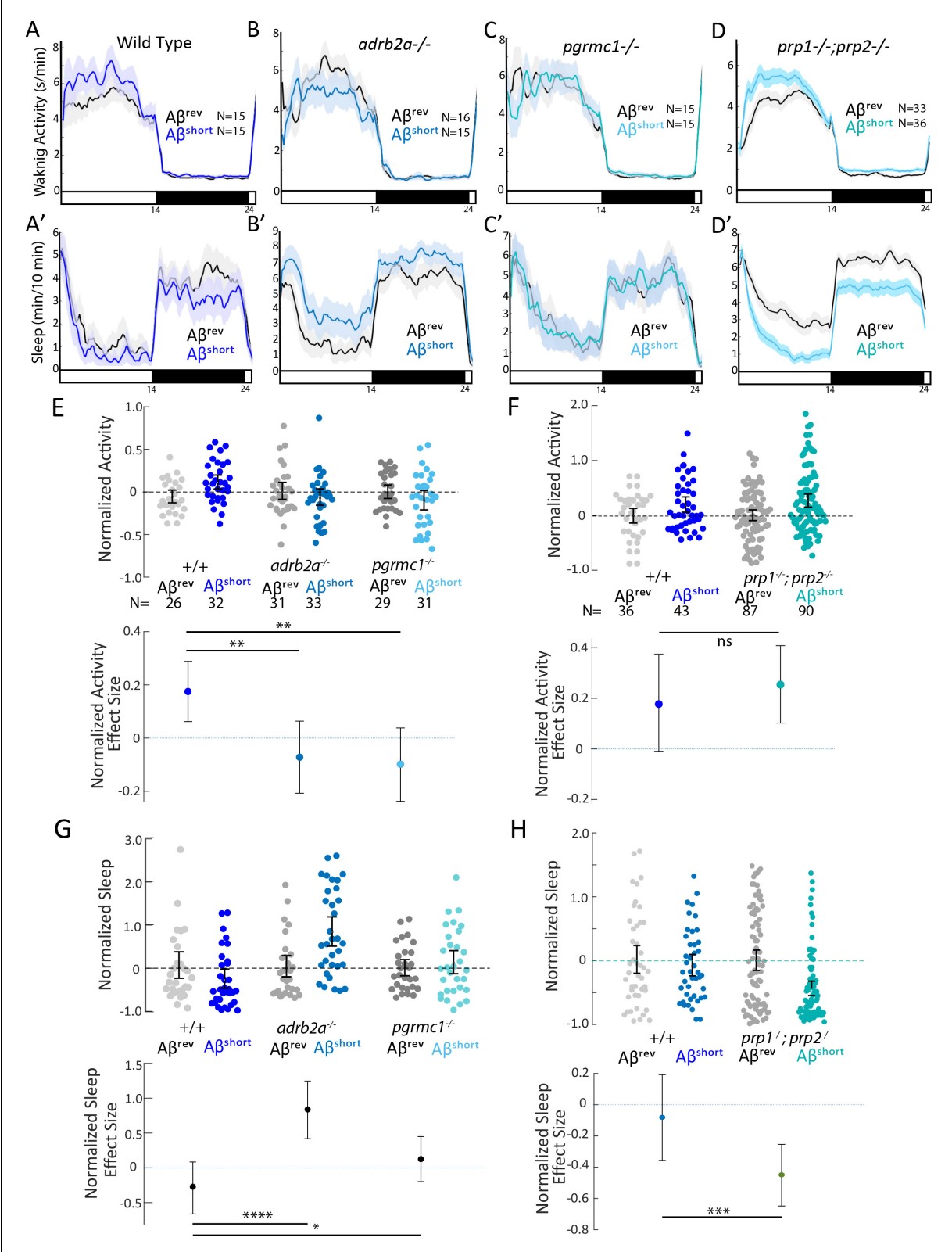

**Figure 3.** Wake induction by Aβ[short] requires Adrb2a and Pgrmc1, but not the Prion Protein. (A-D') Exemplar 24 hr traces comparing the effects of Aβ[short] oligomers on average waking activity (A-D) and sleep (A'-D') versus Aβ[rev] injected into wild type (A,A'), *adrb2a-/-* (B,B'), *pgrmc1-/-* (C,C'), and *prp1-/-;prp2-/-* mutants (D,D'). (E-H) The effect of Aβ[short] relative to Aβ[rev] on normalized waking activity (E and F) and sleep (G and H) during the first day is shown. Each dot represents a single larva normalized to the mean of the Aβ[rev] control, and error bars indicate ± SEM. The mean difference effect

*Figure 3 continued on next page*

*Figure 3 continued*

size and 95% confidence interval are plotted below. N = the number of larvae. The wake inducing and sleep suppressing effects of Aβ$^{short}$ are absent in (**E,G**) *adrb2a-/-* and *pgrmc1-/-* but enhanced in *prp1-/-;prp2-/-* mutants (**F,H**). $^{ns}p>0.05$, $^{*}p\leq0.05$, $^{**}p\leq0.01$, $^{***}p\leq0.0001$, $^{****}p\leq10{-}5$ one-way ANOVA. Data is pooled from n = 2 independent experiments for *adrb2a-/-*and *pgrmc1-/-* and n = 3 for *prp1-/-;prp2-/-*. See also *Figure 3—figure supplements 1* and *2*.

The online version of this article includes the following figure supplement(s) for figure 3:

**Figure supplement 1.** Crispr/Cas9 targeting of zebrafish *adrb2a* and *pgrmc1*.

**Figure supplement 2.** *adrb2a* and *pgrmc1* mutations have small effects on baseline sleep:wake parameters.

exposure to Aβ$^{long}$. Overall, these results are consistent with the effect of genetic ablation of *prp1* and *prp2*. Thus, both genetic and pharmacological interference with several steps of the Aβ-Prp-mGluR5-Fyn kinase signaling cascade prevents the ability of Aβ$^{long}$ to increase sleep behavior.

## Aβ$^{short}$ and Aβ$^{long}$ affect sleep through distinct neuronal/molecular pathways

Although long and short oligomers require different receptors to affect behavior, they may act within the same neuronal circuit signalling cascade. If so, one phenotype should predominate over the other when the two oligomers are co-administered. Alternatively, oligomers may signal through parallel signalling circuits to bi-directionally modulate sleep in an additive manner. To test this, we co-injected Aβ$^{short}$ and Aβ$^{long}$ in a 1:1 ratio and compared the sleep phenotype to injection of either oligomer alone. As expected, Aβ$^{short}$ alone reduced sleep and Aβ$^{long}$ alone increased sleep. In contrast, co-injection of both Aβ$^{short}$ and Aβ$^{long}$ resulted in an intermediate phenotype that is indistinguishable from control injections of Aβ$^{rev}$ (*Figure 6F*), suggesting that the effects of Aβ$^{short}$ and Aβ$^{long}$ are additive and likely act through distinct neuronal circuits or signaling cascades to modulate sleep.

Considering this result is aligned with the genetic and pharmacological data, we propose a bi-directional model of Aβ sleep regulation in which Aβ$^{short}$ and Aβ$^{long}$ act through distinct receptors and neuronal pathways to independently modulate behavioral state (*Figure 6G*). In this model, the presence of both oligomers leads a balance of signaling through sleep-promoting and sleep-inhibiting pathways, resulting in little or no change in behavior. Tipping the balance from one oligomeric state to the other leads to either the sleep activating or the sleep inhibiting pathway to predominate.

## Discussion

Previous studies have suggested that changes in sleep during AD may further accelerate Aβ accumulation and neuronal damage, creating a vicious cycle that leads to further neuronal dysregulation and increased sleep-wake cycle abnormalities (*Roh et al., 2012*). Our results show that, depending on its oligomeric form, Aβ can acutely increase or decrease sleep and wake behaviors and brain states through behaviorally relevant molecular targets and independently of neuronal cell death. The exogenous application of Aβ oligomers in our experiments limit the conclusions we can draw about endogenous functions of Aβ, which in vivo may present with different structure, local concentrations, and kinetics. However, the bi-directional Aβ modulation of sleep and wakefulness (*Figure 6G*) predicts that alterations to the relative concentrations of different Aβ oligomeric forms during healthy aging and AD disease progression will have opposing consequences on sleep and wake behavior.

### Distinct molecular pathways for Aβ sleep-wake regulation

We found that Aβ$^{short}$-triggered wakefulness required intact Adrb2a and Pgrmc1, while Aβ$^{long}$-induced sleep required functional Prp signalling. These data are consistent with a model in which Aβ directly binds to these targets to modulate downstream signaling cascades that ultimately affect neuronal circuits that regulate behavioral state. Our results match well with previous reports demonstrating binding preferences of Aβ dimers, trimers, and 56 kDa oligomers for different targets in vitro. For example, Aβ dimers, which have been detected in the brains of AD patients ( *Vázquez de la Torre et al., 2018*), have been shown to directly bind human ADRB2 with high-affinity, causing

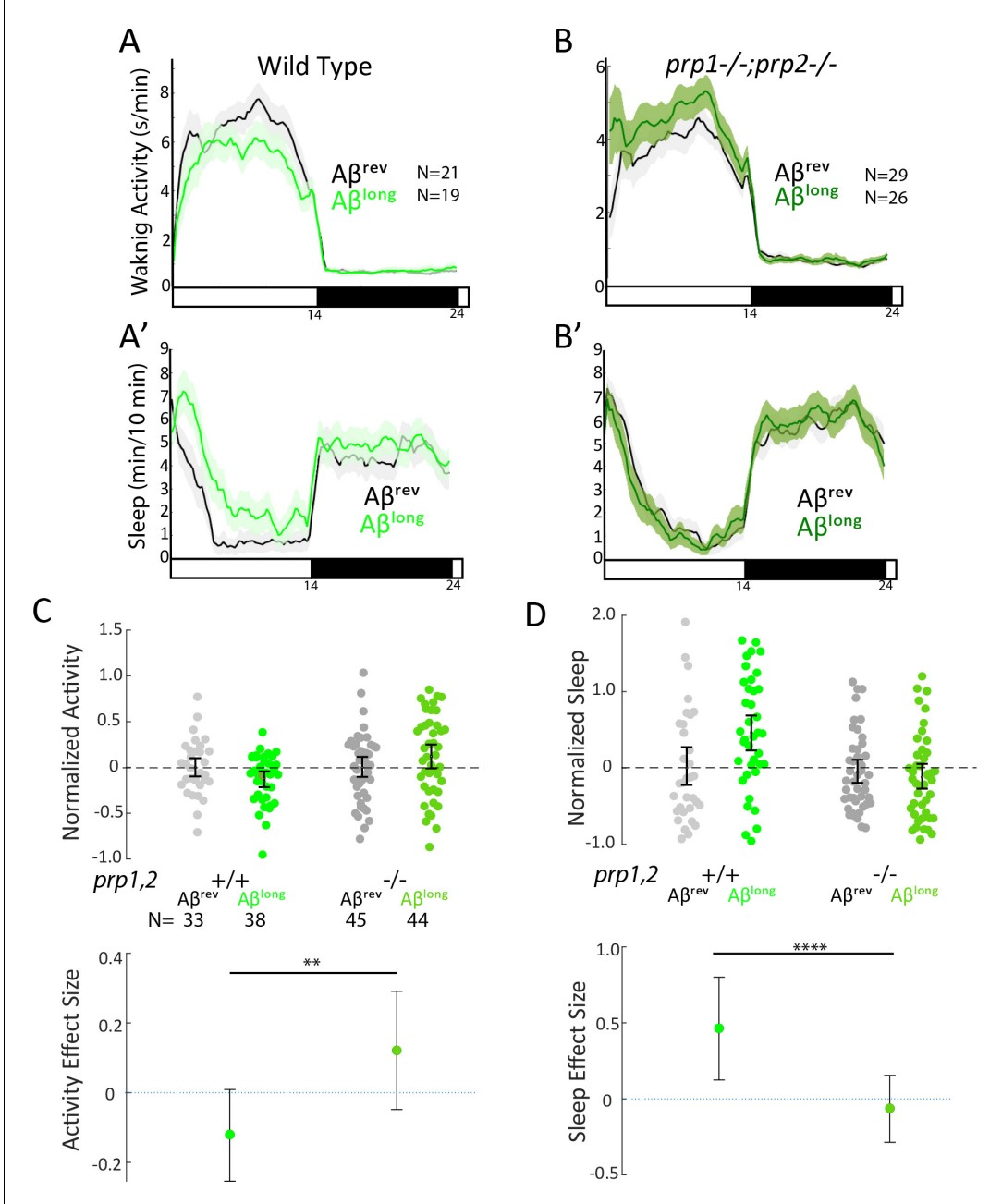

**Figure 4.** Sleep induction by Aβ$^{long}$ requires signalling through Prion Protein. (**A-B'**) Exemplar 24 hr traces comparing the effects of Aβ$^{long}$ oligomers on average waking activity (**A,B**) and sleep (**A'-B'**) versus Aβ$^{rev}$ on wild type (**A,A'**), and *prp1$^{-/-}$;prp2$^{-/-}$* mutant (**B,B'**) backgrounds. (**C-D**) The effect of Aβ$^{long}$ relative to Aβ$^{rev}$ on normalized waking (**C**) and sleep (**D**) on wild type and *prp1$^{-/-}$;prp2$^{-/-}$* mutant backgrounds (mixed *prp3* background) during the first day is shown. The activity reducing (**C**) and sleep promoting (**D**) effects of Aβ$^{long}$ are blocked in *prp1$^{-/-}$;prp2$^{-/-}$* mutants. **p≤0.01, ****p≤10–5 one-way ANOVA. Data is pooled from n = 3 independent experiments. See also *Figure 4—figure supplements 1* and *2*.

The online version of this article includes the following figure supplement(s) for figure 4:

**Figure supplement 1.** Relationship among zebrafish *prp* genes with Aβ binding sites.

**Figure supplement 2.** *prp* double mutants do not affect baseline sleep or wake across the day:night cycle.

increased calcium influx and neuronal hyper-activation in rat prefrontal cortical slices (*Wang et al., 2010*). PGRMC1 can be activated by AD brain extracts (*Izzo et al., 2014b*) and also has shown preferential binding for 50–75 kDa Aβ species in vitro (*Izzo et al., 2014a*). Both types of shorter Aβ species that bind to ADRB2 and PGRMC1 fall within the size ranges that induce Adrb2a- and Pgrmc1-

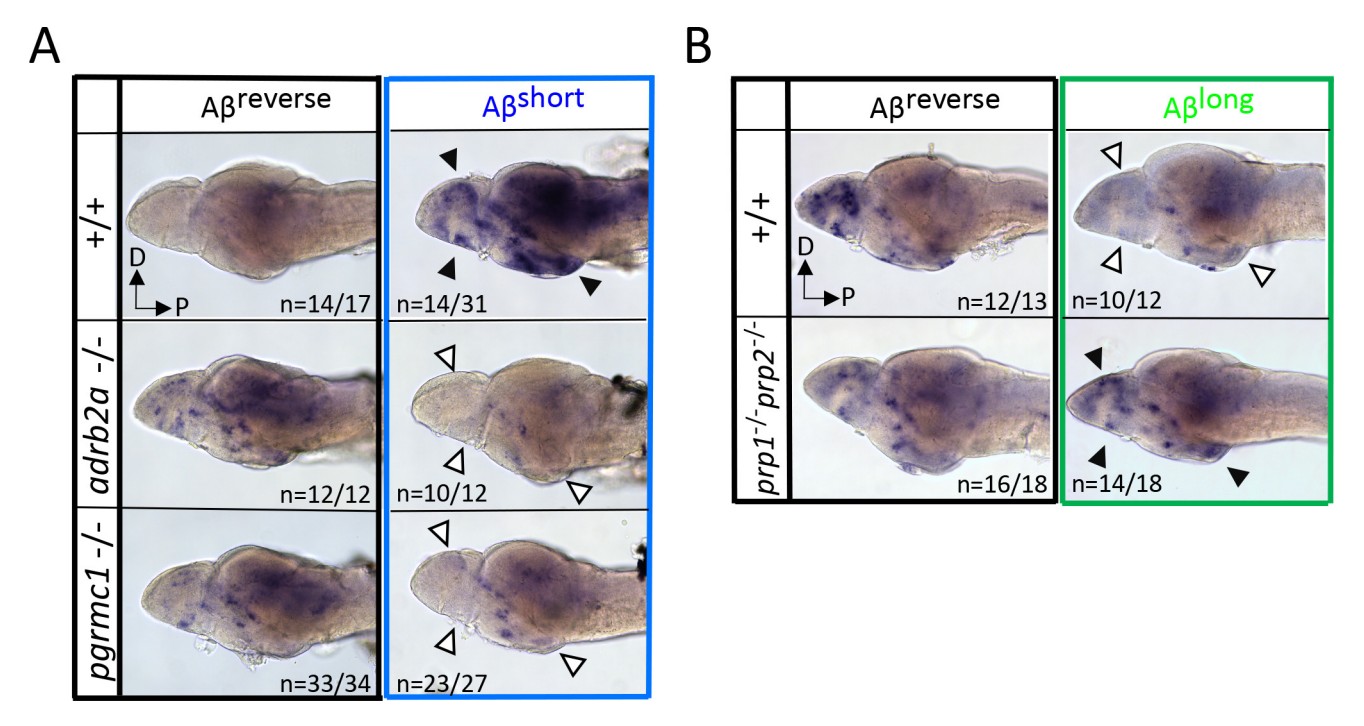

**Figure 5.** Neuronal activity after exposure to Aβ preparations is altered in mutants of Aβ binding targets. (**A**) After Aβ$^{short}$ injection into WT larvae (top right), *c-fos* is detected in many larval brain areas, including the dorsal and ventral telencephalon and posterior hypothalamus (black arrowheads), but not after injection of Aβ reverse controls (left). In contrast, Aβ$^{short}$ injections into either *adrb2a*$^{-/-}$ (middle right) or *pgrmc1*$^{-/-}$ mutants (bottom right) do not induce *c-fos* expression. The brains in the middle and lower panels were stained longer than the WT (+/+) brains to ensure detection of weaker expression. D = dorsal, p=Posterior. n = blinded counts of brains with expression pattern/total brains. (**B**) Compared to Aβ$^{rev}$ injections, Aβ$^{long}$ oligomers induce less *c-fos* expression in WT larvae (top panels). In contrast, Aβ$^{long}$ induced relatively increased *c-fos* expression in the telencephalon and posterior hypothalamus (black arrows) in the *prp1*$^{-/-;}$ *prp2*$^{-/-}$ double mutants. These Aβ$^{rev}$ and Aβ$^{long}$injectedbrains were stained longer to ensure detection of weaker *c-fos* expression. D = dorsal, p=Posterior.

dependent wakefulness in zebrafish larvae. Our results are also consistent with studies that have identified PrP as a direct binding partner (*Laurén et al., 2009*) for longer Aβ-oligomers of 20–200 nm in length (*Nicoll et al., 2013*), the size range of our sleep-inducing Aβ$^{long}$ preparation. In neuronal culture and slice preparations, longer Aβ-oligomers trigger reduction of synaptic strength (*Laurén et al., 2009*) via a Prp signaling cascade through mGluR5 and Fyn kinase activation (*Um et al., 2012*). Similarly, we found that pharmacological blockade of either the direct Aβ-Prp interaction (with Chicago Sky Blue 6B), mGluR5 signalling (with MPEP), Fyn kinase activity (with saracatinib), or by mutation of the Prp receptors prevented the widespread reduction of neuronal activity and increase in sleep that was induced by longer Aβ-oligomers.

Although triggering neuronal and behavioral changes through distinct molecular pathways, several aspects of Aβ's effects on sleep-wake regulation remain to be elucidated. For example, the elimination of either Adrb2a or Pgrmc1 is sufficient to fully prevent Aβ$^{short}$-induced wakefulness. This suggests that Adrb2a and Pgrmc1 function in the same molecular pathway, and signaling by Aβ on either alone is insufficient to modulate behavior. Not much is known about how these two receptors interact with one another, but at least one study (*Roy et al., 2013*) has suggested they can directly physically interact. Whether Aβ$^{short}$ binds both receptors to affect behavior or whether one receptor is an obligate component of the other's ability to transmit Aβ signals is currently unclear. It also remains unclear if the Aβ$^{short}$-Adrb2a/Pgrmc1 wake pathway and the Aβ$^{long}$-Prp sleep pathway occur in the same or different sets of neurons to modulate behavior, as these receptors have widespread expression in zebrafish (*Cotto et al., 2005*; *Málaga-Trillo et al., 2009*; *Steele et al., 2011*; *Thisse and Thisse, 2004*; *Wang et al., 2009*), although our co-injection experiment suggests they act on parallel neuronal circuits. Numerous wake- and sleep-promoting neuronal populations that

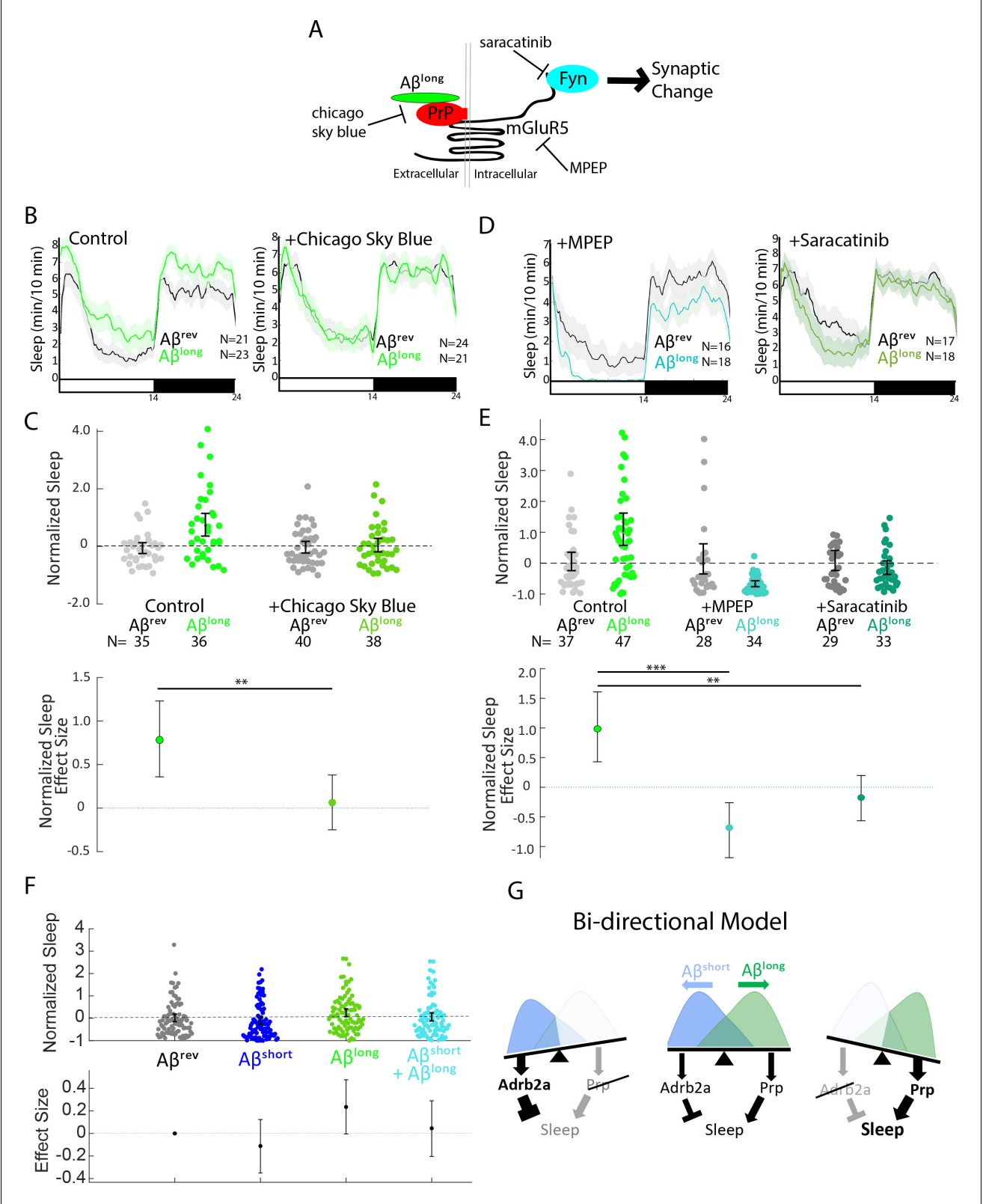

**Figure 6.** Pharmacological blockade of the Aβ$^{long}$-Prp-mGluR5-Fyn Kinase signaling cascade prevents increases in sleep. (**A**) Schematic showing how Aβ–Prp interactions signal through mGluR5 to activate Fyn kinase, leading to synaptic changes (***Nygaard et al., 2014***). Small molecules that block each step in the pathway are indicated. (**B**) Representative traces of sleep behavior after Aβ$^{long}$ versus Aβ$^{rev}$ injections in the absence (left) or presence (right) of the Aβ-Prion binding disruptor, Chicago Sky Blue 6B (3 nM). Ribbons represent ± SEM. (**C**) The effect of Aβ$^{long}$ relative to Aβ$^{rev}$ on normalized sleep

*Figure 6 continued on next page*

Figure 6 continued

during the first day in the in the absence or presence of 3 nM Chicago Sky Blue 6B. The data is pooled from n = 2 independent experiments **p≤0.01, one-way ANOVA. (D) Representative traces of sleep behavior after Aβ$^{long}$ versus Aβ$^{rev}$ injections in the presence of mGluR5 inhibitor MPEP (5 uM, left) and Fyn Kinase inhibitor saracatinib (300 nM, right). Ribbons represent ± SEM. (E) The effect of Aβ$^{long}$ relative to Aβ$^{rev}$ on normalized sleep during the first day in the absence or presence of 5 uM MPEP (left) and 300 nM saracatinib (right). Each dot represents a single larva normalized to the mean Aβ$^{rev}$. Data is pooled from two independent experiments. **p≤0.01, ****p≤10–5 one-way ANOVA. (F) The effect of a 1:1 mixture of Aβ$^{long}$ to Aβ$^{short}$ relative to single injections of Aβ$^{rev}$, Aβ$^{short}$, and Aβ$^{long}$ on normalized sleep during the first day. The data is pooled from n = 4 independent experiments. (G) A bi-directional model for sleep/wake regulation by Aβ. In wild-type animals (centre), injection of Aβ$^{short}$ species signal through Adrb2a/Pgrmc1 to drive wakefulness while Aβ$^{long}$ oligomers signal via Prp to induce sleep. In mutants that lack Prp (left), only Aβ$^{short}$ species (as shown by the overlapping distributions) remain to inhibit sleep with no residual Aβ$^{long}$ oligomers to stimulate the sleep-inducing pathway to counteract wake-inducing signals. Thus prp1$^{-/-}$; prp2$^{-/-}$ mutants have enhanced wakefulness in response to Aβ. Conversely, mutants that lack Adrb2a/Pgrmc1 (right), retain only the sleep-promoting Aβ pathway and fail to increase wakefulness in response to Aβ$^{short}$. See also *Figure 6—figure supplement 1*.

The online version of this article includes the following figure supplement(s) for figure 6:

**Figure supplement 1.** Pharmacological blockade of the Aβ$^{long}$-Prp-MgluR5-Fyn Kinase signalling cascade prevents reductions in waking activity.

could serve as neuronal targets for these signalling cascades to drive changes in behavioral state have been uncovered in zebrafish (*Barlow and Rihel, 2017*). Future experiments will be needed to tease out the neuronal components involved, for example by replacing functional receptors into candidate neurons in otherwise mutant animals and rescuing the responses to Aβ, or by mutating receptors selectively in cell types with conditional genetics.

## Altered Aβ oligomeric ratio—implications for sleep in health and AD

Our model investigates alterations in sleep/wake behavior due to acute changes in exogenously delivered Aβ levels. Thus, it is possible that the sleep/wake effects observed in our study may be different from those observed when Aβ fluctuates over 24 hr or when it is chronically accumulating as in AD. However, our model predicts that alterations to the ratio of Aβ oligomeric forms present in the brain could have differential effects on sleep-wake regulation, as the balance between sleep- and wake- promoting Aβ signals is tilted to favour one pathway over the other (*Figure 6G*).

Given the natural daily increase in Aβ secretion during wakefulness and increased levels of Aβ clearance during sleep (*Xie et al., 2013*), changes in extracellular Aβ levels could sculpt behavior over the normal 24 hr circadian cycle. As Aβ burden is acutely increased by sleep deprivation (*Shokri-Kojori et al., 2018*), perturbations to the normal sleep-wake cycle may feedback on behavior through altered Aβ signaling. Other phenomena have been reported to alter Aβ generation and fibrilization over short time-scales. For example, temperature changes in the physiological range (35–42℃) have been reported to significantly affect Aβ oligomerization (*Ghavami et al., 2013*), suggesting that either the natural daily fluctuation in body temperature (in humans, up to 2℃) or the induction of a fever can promote changes in amyloidogenic Aβ generation (*Szaruga et al., 2017*). In addition, Aβ can act as an antimicrobial peptide (*Kumar et al., 2016*; *Soscia et al., 2010*), and microbial infection can trigger Aβ fibrilization (*Eimer et al., 2018*). Considering that infection and fever are also potent drivers of sleep (*Imeri and Opp, 2009*), the sleep-inducing Aβ-Prp signaling pathway we identified here could mediate recovery sleep during illness– a hypothesis for future investigation.

On longer timescales, the amount and type of Aβ oligomeric species (including dimers, cross-linked dimers, trimers, and 56 kDa oligomers) found in healthy brains change across the human life cycle (*Lesné et al., 2013*) and are heterogeneous and elevated in AD patients (*Izzo et al., 2014a*; *Kostylev et al., 2015*). Although the precise makeup of Aβ species present in healthy and AD brains has remained difficult to quantify (*Benilova et al., 2012*), some studies have indicated that short (dimers, trimers) Aβ oligomers are more enriched in the early, mild cognitive impairment (MCI) stages of AD, while longer oligomers predominate in the CSF at later clinical stages of AD (*De et al., 2019*). Similarly, AD progression is associated with increasingly large disruptions in sleep patterns, with patients exhibiting high levels of sleep fragmentation, a lack of circadian rhythm, night-time insomnia and irregular daytime napping throughout the day (*Videnovic et al., 2014*). One possibility consistent with our data is that sleep symptoms of both normal aging and AD may reflect changes in Aβ burden that lead to an altered balance in sleep- and wake-promoting signaling

cascades. These signaling molecules might therefore be potential therapeutic targets for treating disrupted sleep early in AD progression, which may in turn slow disease progression.

## Materials and methods

See the Key Resources Table (Appendix 1—key resources table) for details of reagents.

### Zebrafish strains and husbandry

Zebrafish (*Danio rerio*) were raised under standard conditions at 28°C in a 14:10 light:dark cycle and all zebrafish experiments and husbandry followed standard protocols of the UCL Zebrafish Facility. AB, TL and ABxTup wild-type strains were used in this study. *prp1* (ua5003/ua5003), *prp2* (ua5001/5001), *adrb2a* (u511/u511) and *pgrmc1* (u512/u512) mutants were outcrossed multiple times, and *pgrmc1* F2 and *adrb2a* F2 and later generations were used for behavior. Ethical approval for zebrafish experiments was obtained from the Home Office UK under the Animal Scientific Procedures Act 1986 with Project licence numbers 70/7612 and PA8D4D0E5 to JR.

### Aβ preparations

HFIP treated Aβ42 peptide (JPT Peptide Technologies) and Aβ42–1 reversed peptide (Sigma) were dissolved in DMSO, vortexed occasionally for 12 min at room temperature and sonicated for 5 min to obtain 100 µM solution. The stock solutions were aliquoted as 5 µl in individual tubes and are kept are −80°C. 1 µl of the 100 µM stock was diluted in (Phosphate buffered saline) PBS to yield 10 µM solutions which were incubated at 4°C, 25°C or 37°C for 24 hr (*Kusumoto et al., 1998*; *Orbán et al., 2010*; *Whitcomb et al., 2015*). 1 hr before injecting, this stock was diluted to 10 nM using 1:10 serial dilutions in PBS and kept at the respective temperature (4°C, 25°C, 37°C) until injecting.

### Transmission Electron Microscopy (TEM)

1 µl of (4°C, 25°C or 37°C incubated) 10 µM Aβ solution was loaded onto formvar/carbon coated 300 mesh grids from Agar Scientific. The grid was washed twice in 20 mM phosphate buffer for 10 s and negatively stained in 2% aqueous uranyl acetate for 30 s. After drying for 2–3 days, samples were imaged using a Phillips TEM. At least five micrographs were used for each condition to blindly measure the length of the Aβ42 oligomeric structures using at least 30 measurements/condition. Using FIJI, 30–50 measurements were taken for each condition by drawing a free-hand line on the fibril, which was then scaled using the scale bar.

### Heart injections

Injections were carried out blindly with a Pneumatic PicoPump (WPI) and glass capillary needles (Science Products Gmbh) prepared with a Micropipette Puller (Shutter Instruments). Five dpf larvae were anesthetized using 4% Tricaine (42 mg/L, Sigma) 30 min before injections. Larvae were immobilized in 1.8% low melting point agarose (ThermoFischer) in fish water on their sides on a slide. 1 nL of Aβ (10 nM starting concentration) was injected into the heart chamber of the fish along with a high molecular weight fluorescent dye (2000 kDa dextran-conjugated FITC (3 mg/ml, Sigma). We estimate that 1 nl of a 10 nM Aβ injection into a ~ 3.01 (±0.16) x$10^8$ µm$^3$(285–317 nL) 5dpf larva yields a final monomeric brain/CSF concentration of ~28–32 pM. The success of the injection was checked under a standard fluorescent scope by the presence of fluorescence in the heart of the animal. Larvae were transferred to fresh fish water for 20 min to recover from Tricaine and transferred to sleep/wake behavior box. For drug blocking experiments, zebrafish larvae were soaked into 3 nM Chicago Sky Blue 6B (Sigma), 5 µM MPEP (Cambridge Biosciences), or 300 nM Saracatinib (Generon) 1 day before the injections (from 4 to 5 dpf). Fluorescently tagged HiLyte Fluor 647-labeled Aβ42 (Eurogentech LTD) was injected at 10 µM.

### Behavioral experiments

Larval zebrafish behavioral experiments were performed and analysed as described (*Rihel et al., 2010*). Briefly, 5 dpf larvae were transferred to 96 square-well plates and continuously illuminated with IR and white lights from 9 am to 11 pm in a Zebrabox (Viewpoint life sciences) for 24–48 hr. The

movement of each larva was measured and duration of movement was recorded with an integration time of 10 s. Data were processed according to *Rihel et al., 2010*, and statistical tests were performed using MATLAB. Mutant larval zebrafish experiments were performed on siblings from heterozygous in-crosses, differing only in the mutation of the specific gene and genotyped at the end of the experiment.

## Dark pulse experiments

Larvae were placed in the behavior tracking boxes, and two or three dark pulses for 10 min with a 2–4 hr interval were introduced in four independent experiments. For data analysis, only the dark pulses after the acclimatization period in the late afternoon were combined for each genotype.

## In situ hybridization

RNA in situ hybridization (ISH) to detect *c-fos* and *galanin* was performed as described (*Thisse and Thisse, 2008*). Zebrafish larvae were fixed in 4% paraformaldehyde in PBS at 4°C overnight. A template for in vitro transcription was generated by PCR using a reverse primer that contains a T7 promoter sequence 5'-TAATACGACTCACTATAGGG-3' from cDNA. A digoxygenin (DIG)-labelled antisense RNA probe was synthesized using the DIG labelling kit (Roche) and T7 RNA polymerase according to the manufacturer's recommendations. The probe was detected with anti-DIG-AP antibody (1:2,000, Roche) and nitro-blue tetrazolium chloride (NBT)/5-bromo-4-chloro-3'-indolyphosphate (BCIP) substrate (Roche) according to published protocols. To detect the differences in expression between the mutant backgrounds and WTs after Aβ$^{short}$ and Aβ$^{long}$ injection, larvae were incubated and washed in the same tubes throughout the ISH procedure to avoid staining artefacts. To do this, larvae from different genotypes were marked by cutting the tail to allow identification after ISH. The brains were exposed by dissection, keeping the brain and spinal cord intact. Embryos were stored in 60% glycerol/PBS for imaging.

## Baseline c-fos ISH

Zebrafish larval siblings were kept in 14:10 day/night normal or reverse-cycle incubators. 50 larvae were collected at each time point (ZT1, ZT13, and ZT19) and fixed in 4% paraformaldehyde in PBS overnight. RNA in situ hybridization (ISH) to detect *c-fos* was performed as described (*Thisse and Thisse, 2008*).

## KASP genotyping

For rapid genotyping of mutant zebrafish harbouring the *adrb2aΔ8* and *pgrmc1Δ16* alleles, a mutant allele-specific forward primer, a wild-type allele-specific forward primer and a common reverse primer were used (LGC Genomics). The primer sequences were targeted against the following:

adrb2a
5'TTTTACTACTTACTGTTTGCACAAACCTATGTTAACTGTGTTAACGTGTGTTTTCTTCTGCTTTTC
TTTCTTGATCTCTGTCAGGTCATGGGAAACATAAGGTCCTCAATACC[**CGAAGATC/-**]TTATCTG
TCCAAACAATACTAATGCCTCCACCAAAAGCGAACTACAGATGACAGTGCTGGGCACACTCA
TGTCCATTCTTGTCTTGATCATCGTCTTTGGCAATGTGATGGTGATTACAGCCA-3'
pgrmc1
5'ATGGCTGAAGAAGCAGTCGAGCAAACTTCTGGAATCCTTCAGGAAATTTTCACGTCGCCAC
TGAACATCAGTTTGCTATGTCTTTGTTTGTTCCTACTTTACAAAATCATCCGCGGAGACAAGCC
[**TGCAGACTATGGCCCG/-**]GYTGAGGAGCCGCTGCCCAAACTCAAGAAAAGAGATTTYAC
TTTAGCAGATCTGCAAGAGTACGATGGACTGAAAAACCCAAGAATCCTGATGGCTG
TCAACGGG-3'

where [x/-] indicates the indel difference in [WT/mutant]. PCR amplification was performed using KASP Master mix (LGC Genomics) according to the manufacturer's instructions. Fluorescence was read on a CFX96 Touch Real-Time PCR Detection System (Bio-Rad) and the allelic discrimination plot generated using Bio-Rad CFX Manager Software.

## Time lapse confocal microscopy

Three Casper larvae at five dpf were mounted dorsally on a slide in 1.5% agarose. 10 nl of Aβ42-Hi488 was intra-cardiac injected to embryos, control fish were untreated. Fish were imaged for 6 min

taking 2-µm-thick stacks through the whole brain using a confocal microscope (Leica SP8). Each fish was imaged every 20 min for 8 hr.

## pERK/tERK staining and activity Mapping

Larvae were fixed overnight at 4°C in 4% paraformaldehyde (PFA) and 4% sucrose in PBS; permeabilized 45 min in 0.05% trypsin-EDTA on ice; blocked 6 hr at room temperature (RT) in phosphate buffered saline plus 0.05% Triton (PBT) plus 2% normal goat serum, 1% BSA, and 1% DMSO; and then incubated over sequential nights at 4°C in primary antibodies (Cell Signaling Technology 4370 and 4696; 1:500) and secondary antibodies conjugated with Alexa fluorophores (Life Technologies; 1:200) in PBT plus 1% BSA and 1% DMSO.

Larvae were mounted in 1.5% low melt agarose and imaged with a custom two-photon microscope (Bruker; Prairie View software) with a 203 water immersion objective (Olympus).

Images were noise filtered using a custom MATLAB (The MathWorks) scripts and registered into Z-Brain using the Computational Morphometry Toolkit (http://www.nitrc.org/projects/cmtk/) with the command string: -a -w -r 0102 l af -X 52 C 8 G 80 R 3 -A '–accuracy 0.4 –auto-multi-levels 4' -W '–accuracy 1.6' -T 4. Registered images were prepared using a custom MATLAB/MIJ (http://bigwww.epfl.ch/sage/soft/mij/) script to downsize, blur, and adjust the maximum brightness of each stack to the top 0.1% of pixel intensities to preserve dynamic range. Activity maps were generated using MATLAB scripts (*Randlett et al., 2015*).

## Crispr/Cas9 mutant generation

The CRISPR design tool CHOPCHOP (http://chopchop.cbu.uib.no) was used to identify a target region in zebrafish *adrb2a* and *pgrmc1* (*Labun et al., 2019*). The gene-specific oligomers were ordered from Thermofisher including the 5' and 3' tags:

> For *adrb2a*:
> 5'ATTTAGGTGACACTATAGTTTGGACAGATAAGATCTTGTTTTAGAGCTAGAAATAGCAAG-3'
> For *pgrmc1*:
> 5'ATTTAGGTGACACTATATGCAGACTATGGCCCGGTTGGTTTTAGAGCTAGAAATAGCAAG-3'
> Constant oligomer:
> 5'AAAAGCACCGACTCGGTGCCACTTTTTCAAGTTGATAACGGACTAGCCTTATTTTAACTTGC TATTTCT AGCTCTAAAAC-3'

The constant oligomer and the gene-specific oligomer were annealed on a PCR machine and filled in using T4 DNA polymerase (NEB) (*Gagnon et al., 2014*). The template was cleaned up using a PCR clean-up column (Qiaquick) and the product was verified on a 2% agarose gel. The sgRNA was transcribed from this DNA template using Ambion MEGAscript SP6 kit (*Gagnon et al., 2014*). Cas9 mRNA and the purified sgRNA were co-injected into one-cell stage embryos at a concentration of 200 ng and 100 ng per embryo, respectively.

## Caspase-3 staining

5dpf larvae that were injected with Aβ oligomers or soaked in Camptothecin (1 µM; Sigma Aldrich) were fixed 5 hr after injection/drug treatment in 4% PFA and kept overnight at 4°C. Brains were dissected and dehydrated the next day and were washed three times in PDT buffer (0.3 Triton-X in PBST with 1% DMSO) and incubated with Caspase-3 antibody (1:500; BD Biosciences) at 4°C. The brains were incubated with Alexa Fluor 568 goat anti-rabbit antibody (1:200; Invitrogen) next day at 4°C overnight and imaged using a confocal microscope.

## Statistical analyses

For data analyses, we used a Gardner-Altman estimation plot, which visualizes the effect size and displays an experimental dataset's complete statistical information (*Ho et al., 2019*). The bootstrapped 95% confidence interval (CI) was calculated from 10,000 bootstrapped resamples (*Ho et al., 2019*).

Details of statistics used in each panel are also described in the figure legend. For multiple comparisons, data was first tested for normality by the Kogloromov-Smirnov (KS) statistic and extreme outliers were discarded by Grubb's test (p≤0.01). Those that violated normality were analysed with the non-parametric Kruskal-Wallis test, with either Tukey-Kramer or Bonferroni post hoc testing;

otherwise, data was analysed with one-way ANOVA followed by Tukey's post-hoc testing. For dose response curves, a two-way ANOVA was performed to test the interaction effects between preparation type and dose. For return to baseline statistics, paired t-tests were performed. Survival curves were analysed with Kaplan-Meier log rank test. All statistical tests and graphs were generated in MATLAB (R2015a) loaded with the Statistical Toolbox. Injection and tracking experiments were performed blinded.

## Acknowledgements

We thank Tom Hawkins and Mark Turmaine for assistance with TEM, Gaia Gestri for assistance with heart injections and Marcus Ghosh for assistance with tERK/pERK experiments. We also thank Dervis Salih, Steve Wilson, and John Hardy for their comments and all first-floor fish lab members for their input throughout the project.

## Additional information

### Funding

| Funder | Grant reference number | Author |
| --- | --- | --- |
| University College London | Excellence Fellowship | Jason Rihel |
| European Research Council | 282027 | Jason Rihel |
| Alzheimer's Research UK | | Jason Rihel |
| Alzheimer Society of Alberta and Northwest Territories | | W Ted Allison |
| Alberta Prion Research Institute of Alberta Innovates | | W Ted Allison |
| Alzheimer Society of Canada | | Patricia LA Leighton |
| Alberta Innovates | | Patricia LA Leighton |
| Wellcome | 217150/Z/19/Z | Jason Rihel |

The funders had no role in study design, data collection and interpretation, or the decision to submit the work for publication.

### Author contributions

Güliz Gürel Özcan, Conceptualization, Formal analysis, Methodology, Writing - original draft, Writing - review and editing; Sumi Lim, Formal analysis, Methodology, Writing - original draft; Patricia LA Leighton, W Ted Allison, Resources, Writing - review and editing; Jason Rihel, Conceptualization, Resources, Formal analysis, Supervision, Funding acquisition, Methodology, Writing - original draft, Project administration, Writing - review and editing

### Author ORCIDs

W Ted Allison (ID) https://orcid.org/0000-0002-8461-4864
Jason Rihel (ID) https://orcid.org/0000-0003-4067-2066

### Ethics

Animal experimentation: Ethical approval for zebrafish experiments was obtained from the Home Office UK under the Animal Scientific Procedures Act 1986 with Project licence numbers 70/7612 and PA8D4D0E5 to JR.

### Decision letter and Author response

Decision letter https://doi.org/10.7554/eLife.53995.sa1
Author response https://doi.org/10.7554/eLife.53995.sa2

## Additional files

### Supplementary files

- Transparent reporting form

### Data availability

All data generated or analyzsed during this study are included in the manuscript and supporting files.

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

# Appendix 1

**Appendix 1—key resources table**

| Reagent type (species) or resource | Designation | Source or reference | Identifiers | Additional information |
|---|---|---|---|---|
| Gene (*Danio rerio*) | *prp1* | *Leighton et al., 2018* PMID:29903907 | ZFIN ID: ZDB-GENE-041221-2 | |
| Gene (*Danio rerio*) | prp2 | *Fleisch et al., 2013* PMID:23523635 | ZFIN ID: ZDB-GENE-041221-3 | |
| Gene (*Danio rerio*) | adrb2a | This paper | ZFIN ID: ZDB-GENE-100414-3 | |
| Gene (*Danio rerio*) | pgrmc1 | This paper | ZFIN ID: ZDB-GENE-041114-91 | |
| Strain background (*Danio rerio*) | AB | UCL Fish Facility | | |
| Strain background (*Danio rerio*) | TL | UCL Fish Facility | | |
| Strain background (*Danio rerio*) | ABxTup | UCL Fish Facility | | |
| Strain (*Danio rerio*) | *prp1* (ua5003/ua5003) mutant | *Leighton et al., 2018* PMID:29903907 | ZFIN ID: ZDB-ALT-181113-1 | |
| Strain (*Danio rerio*) | *prp2* (ua5001/5001) mutant | *Leighton et al., 2018* PMID:29903907 | ZFIN ID: ZDB-ALT-130724-2 | |
| Strain (*Danio rerio*) | adrb2a (u511/u511) mutant | This paper | | Allele will be added to ZFIN upon publication acceptance |
| Strain (*Danio rerio*) | pgrmc1 (u512/u512) mutant | This paper | | Allele will be added to ZFIN upon publication acceptance |
| Antibody | anti-DIG-AP antibody (Sheep) polyclonal | Roche | Cat # 14608125; RRID:AB_2734716 | (1:2000) |
| Antibody | anti-Active Caspase 3 (Rabbit) | BD Biosciences | Cat # 559565; RRID:AB_397274 | (1:500) |
| Antibody | p44/42 MAP Kinase (L34F12) Mouse mAb | Cell Signaling | Cat # 4696; RRID:AB_390780 | (1:500) |

*Appendix 1—key resources table continued*

| Reagent type (species) or resource | Designation | Source or reference | Identifiers | Additional information |
|---|---|---|---|---|
| Antibody | Phospho-p44/42 MAPK (Erk1/2)(Thr202/Tyr204) Rabbit mAb | Cell Signaling | Cat # 4370; RRID:AB_2315112 | (1:500) |
| Antibody | Alexa Fluor 568 goat anti-mouse, polyclonal | Thermo Fisher Scientific | Cat # A-11031; RRID:AB_144696 | (1:200) |
| Antibody | Goat anti-Rabbit IgG Alexa 488, polyclonal | Thermo Fisher Scientific | Cat # A-11034; RRID:AB_2576217 | (1:200) |
| Sequence-based reagent | *galanin* probe | **Chen et al., 2017** PMID:28648499 | Plasmid for galanin ISH riboprobe | |
| Sequence-based reagent | *c-fos* probe | **Reichert et al., 2019** PMID:31537465 | Plasmid for c-fos ISH riboprobe | |
| Sequence-based reagent | *adrb2a* | This paper | Gene-specific oligomer for CRISPR | 5'ATTTAGGTGACAC TATAGTTTGGA CAGATAAGATCTTG TTTTAG AGCTAGAAATAG-CAAG-3' |
| Sequence-based reagent | *pgrmc1* | This paper | Gene-specific oligomer for CRISPR | 5'ATTTAGGTGACAC TATATGCAGACTA TGGCCCGGTTGG TTTTAG AGCTAGAAATAG-CAAG-3' |
| Sequence-based reagent | gRNA constant region | Thermofisher | Constant oligomer for CRISPR | 5'AAAAGCACCGAC TCGGTGCCACTTTT TCAAGTTGATAACG-GACTAGCCTTA TTTTAACTTGCTATTTC T AGCTCTAAAAC-3' |
| Peptide, recombinant protein | Beta-Amyloid (1-42); HFIP treated | JPT Peptide Technologies | Cat# SP-Ab-07_0.5 | |
| Peptide, recombinant protein | Amyloid $\beta$ 42-1 reverse human | Sigma Aldrich | Cat# SCP0048 | |
| Peptide, recombinant protein | ß-Amyloid (1-42), HiLyte Fluor 647-labeled | Eurogentech LTD | Cat# AS-64161 | |
| Commercial assay or kit | T4 DNA polymerase | NEB | Cat# M0203S | |
| Commercial assay or kit | PCR clean-up column kit | Qiaquick | Cat# 28104 | |
| Commercial assay or kit | Ambion MEGAscript SP6 kit | Ambion | Cat# AM1330 | |
| Chemical compound, drug | 2000 kDa dextran-conjugated FITC | Sigma Aldrich | Cat# 52471 | 3 mg/ml |

*Appendix 1—key resources table continued*

| Reagent type (species) or resource | Designation | Source or reference | Identifiers | Additional information |
|---|---|---|---|---|
| Chemical compound, drug | Chicago Sky Blue 6B | Sigma Aldrich | Cat# C8679 | 3 nM |
| Chemical compound, drug | MPEP | Cambridge Biosciences | Cat# CAY14536 | 5 $\mu$M |
| Chemical compound, drug | Saracatinib | Generon | Cat# A2133 | 300 nM |
| Chemical compound, drug | Pentylenetetrazol (PTZ) | Sigma Aldrich | Cat# P6500 | 10 mM |
| Chemical compound, drug | Camptothecin | Sigma Aldrich | Cat# 208925 | 1 uM |
| Software, algorithm | Sleep analysis2 | *Rihel et al., 2010* PMID:21111222 | | |
| Software, algorithm | Dabest estimation plots | *Ho et al., 2019* PMID:31217592 | | |

