## [Decision Letter]

**Acceptance summary:**

This manuscript provides new insight into mechanisms by which oligomers of amyloid beta might affect sleep. While Alzheimer's Disease is linked to sleep disruption, the nature of the interaction is still unclear. Using a zebrafish model, the authors show that short and long amyloid oligomers interact with different binding partners to decrease or increase sleep respectively.

**Decision letter after peer review:**

Thank you for submitting your article "Bi-directional modification of sleep and wake by amyloid beta oligomers" for consideration by *eLife*. Your article has been reviewed by three peer reviewers, and the evaluation has been overseen by a Reviewing Editor and Michael Eisen as the Senior Editor. The following individual involved in review of your submission has agreed to reveal their identity: Erik Musiek (Reviewer #3).

The reviewers have discussed the reviews with one another and the Reviewing Editor has drafted this decision to help you prepare a revised submission.

The reviewers found the manuscript of interest and considered it a potentially important conceptual advance. However, they had a few concerns, which would need to be addressed for further consideration at *eLife*:

The model is based on intracardiac injection of Aβ, so the phenotypes result from exogenous expression/overexpression. Given this, the authors should refrain from drawing conclusions about endogenous Aβ. At the same time, the manuscript would benefit from minimal characterization of the endogenous molecules. For instance, is there a rhythm of Aβ expression over the sleep:wake cycle?

The fish undergo anesthesia and heart perforation and are recorded a few hours later. What are the controls for handling, surgical stress, and confounds of prior anesthesia? On a related note, can the authors exclude toxicity, which could affect motion? They address this point by showing *c-fos* and ERK staining, but many different patterns are observed and none are compared to staining under baseline sleep:wake conditions. It is also disturbing that the *c-fos* expression is so widespread. The reversibility of the effect is important and the role of specific molecules is interesting, but these still do not demonstrate impact on wake or sleep regulation per se.

Given that AD brains likely have oligomers of all sizes, it would be good to know what happens when short and long oligomers are infused together.

Reviewer 1:

There is a growing appreciation about the fundamental bidirectional link between sleep and Alzheimer's disease. Here Rihel and colleagues use a zebrafish model coupled to the injection of amyloid beta oligomers (the initiating pathogenic species for AD) to examine the link between Aβ and sleep. They demonstrate that the length of the oligomers determines whether Aβ induces wake (short Aβ) or sleep (long Aβ), providing novel insights into the role of different forms on sleep/wake. Importantly, they extend their findings to reveal novel molecular insights into the mechanisms into how Aβ exerts these sleep/wake effects. Overall, the findings make an important advance that will be of interest to the broad *eLife* readership.

I have one significant concern relating to claims that these studies reveal novel functions for the endogenous Aβ. A key missing experiment in this regard is manipulation of the endogenous Aβ gene/protein (or even assessment of endogenous Ab) and thus it is unclear if exogenous (intracardiac) injection of Aβ faithfully reproduces how an endogenous neuronal pathway would deliver Aβ in terms of location, local concentrations and kinetics.

I think the findings are significant and important on their own without having to make this claim which in this case is highly speculative. I would suggest either addressing experimentally or rewording and de-emphasizing this point in the text to make clear the speculative possibilities. In any case, these shortcomings should be more forthrightly noted.

Reviewer 2:

The use of zebrafish to investigate the role of beta amyloid polymers on sleep/wake regulation is potentially interesting as AD patients suffer from insomnia. Here Ozcan and colleagues inject oligomers synthesized in vitro into the fish neonate hearts and fish motion was then recorded and used as a proxy for sleep and wake states. The authors found a correlation between the polymer length and the impact on fish motor and brain activity.

While the findings are potentially interesting, several points are unclear if concerning to the reviewer:

1) First, all the experiments and interpretations rely on overexpression of Aβ polymers, there is no description or investigation in this study of the normal baseline of Aβ accumulation in this species. One would expect to see such data in Figure 1 and Figure 1—figure supplement 1 for example. Is there in fish a night vs. day, sleep vs. night rhythm of Aβ accumulation/expression?

2) The fish undergo anesthesia and heart perforation and are recorded a few hours later. How do handling, surgical stress, and confounds of prior anesthesia can be eliminated from "sleep-wake" data interpretation?

3) It is hard for the reader to distinguish a specific effect on sleep/wake. Increased or decreased motion could be due to toxicity or specific stimulation of neuronal circuits due to non-physiological presence of exogenous oligomers. The authors try to tackle this issue with *c-fos* and ERK staining, but Figure 2 shows at least 6 different staining patterns, none of them compared to a sleep/wake baseline of staining. It is quite worrisome to see such a broad over expression of *c-fos* throughout the brain when Aβ is accumulated. Are the fish having a seizure?

Toxicity could lead to reduced motion and even if it's reversible it can still be transient toxicity until oligomers are washed out. Hyperactivity could be due to aspecific overstimulation of neurons as illustrated by *c-fos* and ERK staining.

4) Injections in mutant backgrounds indeed show some specificity in binding/interaction but still it does not demonstrate that the impact is on wake or sleep regulation per se. Again only motion or broad brain staining (at one time point…) are shown. An alternative interpretation is that adrb2a, pgrmc, prp1 can indeed bind Aβ but relay the toxic or aspecific impact of oligomers over expression in a brain that normally does not accumulate such molecules.

This study has the potential to be extremely interesting but many controls and demonstration of endogenous Aβ role on sleep-wake cycle are needed.

Reviewer 3:

The authors report the use of a novel model of intracardiac infusion of Aβ peptides in zebrafish larvae to study the effects of Aβ on sleep and neuronal activity. They provide convincing data that preparations of shorter Aβ oligomers induce neuronal activity and decrease sleep, while longer oligomers suppress neuronal activity and decrease sleep. They then delete known Aβ receptor proteins, and show that the effects of Aβ-short can be block by deletion of Adrb2 and Pgrmc1, while the effects of Aβ-long are blocked by prion protein deletion, or specific drugs. This is a unique system and method for administering Aβ that is quite powerful, and the experiments are rigorous and generally use multiple converging approaches (for instance genetic+pharmacologic) to support their findings. The reversibility of the effect, as well as blockade with specific pharmacological agents suggests that these are not non-specific toxic events. The findings provide a framework with which to potentially test other neurodegenerative proteins (such as a-syn), and to inform similar studies in mammalian systems.

1) While the experiments are well performed and the data intrinsically consistent, the applicability to mammals (and humans) is a consideration. Infusion of Aβ into the heart of larvae is a highly artificial system, and events that occur during sudden changes in Aβ levels may be different that those observed when Aβ is chronically present (as in AD). For example, infusion of Aβ peptide into the brain of mice or rats can induce acute, local neurodegeneration that is not observed in APP transgenic mice with chronically elevated Aβ levels. This is a fundamental shortcoming of the model, and there is little that can be done to address it, but it should be perhaps mentioned in the Discussion.

2) The implications of this bidirectional effect of short and long oligomers for sleep phenotypes in AD are also a bit unclear, as oligomers of all sizes are likely present in AD brain (though perhaps in different ratios as the disease progresses). It would be helpful to determine which pathway is dominant when both short and long oligomers are infused together, perhaps in different ratios. This is the only experiment I would request.

---

## [Author Response]

The reviewers found the manuscript of interest and considered it a potentially important conceptual advance. However, they had a few concerns, which would need to be addressed for further consideration at eLife:The model is based on intracardiac injection of Aβ, so the phenotypes result from exogenous expression/overexpression. Given this, the authors should refrain from drawing conclusions about endogenous Aβ.

We have modified the text throughout to be clearer about this limitation of our methodology. We address this in our detailed responses to each reviewer’s comments.

At the same time, the manuscript would benefit from minimal characterization of the endogenous molecules. For instance, is there a rhythm of Aβ expression over the sleep:wake cycle?

This point is extremely difficult to address in zebrafish currently. To our knowledge, a daily rhythm of Aβ has only been detected in measurement of the ISF/CSF (Holth et al., 2019; Kang et al., 2009; Roh et al., 2012), but this rhythm is not detected in whole tissue homogenates in mammals e.g.(Kang et al., 2009). Since the total CSF in one zebrafish larval brain is only ~6.5 nl (Fame et al., 2016), and typical experiments can only remove 1-2 nl of fluid per animal, it was not feasible to extract CSF from 900-5000 larvae to obtain ~5 μl of CSF for a single timepoint analysis, let alone the numbers that would be needed across 24hr. We therefore attempted to detect Aβ using whole tissue homogenates, but the sensitivity even for this method was not sufficient for us to detect Aβ in fish (Author response image 1). Potentially higher sensitivity kits were to be tried, but the global pandemic prevented those from being delivered before we were ordered shut.

**Author response image 1. sa2fig1:** A Western blot against Aβ42 is unable to detect a band in homogenates from 35-50 zebrafish larvae or from an adult brain.

The fish undergo anesthesia and heart perforation and are recorded a few hours later. What are the controls for handling, surgical stress, and confounds of prior anesthesia?

We address this with further experiments, but first wish to communicate that from our perspective the controls for these elements were already a strength of the original submission.

To control for the effects of both experimental handling artefacts and exogenous protein confounds, we have included in every experiment a reverse Aβ peptide control that is injected into sibling larvae at the same time as the experimental oligomers, under the same handling and surgical stress and injected at the same concentrations to account for generalized response to proteins, and then tracked the reverse peptide injected and oligomer injected larvae side by side in the same video tracking set-up. Thus, the only difference between the controls and experiments shown in every panel is the inclusion Aβ oligomers vs. a control peptide.

Nevertheless, we have added additional experiments with anaesthesia only, mounting only and injection only controls (new Figure 1—figure supplement 1H-I). Our results demonstrate that the initial placement of fish into behavior boxes at the beginning of the video-tracking is responsible for differences between our experimental setup and other tracking experiments (e.g. Figure 1—figure supplement 1J-K) and is not due to heart-injection, anaesthesia, or mounting.

On a related note, can the authors exclude toxicity, which could affect motion? They address this point by showing c-fos and ERK staining, but many different patterns are observed and none are compared to staining under baseline sleep:wake conditions. It is also disturbing that the c-fos expression is so widespread. The reversibility of the effect is important and the role of specific molecules is interesting, but these still do not demonstrate impact on wake or sleep regulation per se.

To exclude toxicity and to demonstrate the effect of Aβ is on bona-fide sleep and wake, we made 4 new experiments. First, we now compare the *c-fos* patterns induced by oligomers to untreated larvae across the sleep:wake cycle and show that the *c-fos* expression pattern of baseline sleep-wake states match the Aβ induced sleep-wake states closely (Figure 2A-C). Second, we show that Aβ engage *galanin* positive neurons, a known key sleep regulator in fish and humans (Reichert et al., 2019) in a manner consistent with inducing normal sleep and wake states (Figure 2F). Third, we show the bout structures of injected fish are indistinguishable from WT fish and are clearly distinct from seizures (Figure 1—figure supplement 3D-E and Figure 1—video 1). Finally, we show that injected fish respond to acute stimuli the same way as their WT siblings, showing they have rapid state reversibility (Figure 1—figure supplement 3C). All of the new 4 experiments support our interpretation that Aβ-induced behavioral states are bona fide sleep/wake states.

Given that AD brains likely have oligomers of all sizes, it would be good to know what happens when short and long oligomers are infused together.

We have now added this experiment to Figure 6F. Because short and long oligomers are additive, we conclude that not only do they act on different receptor molecules but also likely act on separate signalling pathways/neurons (e.g. instead of in series with each other) to affect sleep. We have added some discussion about this point.

Reviewer 1:[…] I have one significant concern relating to claims that these studies reveal novel functions for the endogenous Aβ. A key missing experiment in this regard is manipulation of the endogenous Aβ gene/protein (or even assessment of endogenous Ab) and thus it is unclear if exogenous (intracardiac) injection of Aβ faithfully reproduces how an endogenous neuronal pathway would deliver Aβ in terms of location, local concentrations and kinetics.

We agree that endogenous manipulation of Aβ levels would be ideal, but we know of no experimental methodology that would allow for the acute manipulation of *only* endogenous Aβ that can also control for oligomer type. For example, genetic manipulation of APP, either through mutation or overexpression of multiple, mutated copies as is done in mouse AD research, will also affect the formation of other APP cleavage products, many of which are known to regulate biological processes, and these effects will accumulate over time, with little to no temporal control over the process. This consideration dictated our methodological choice from the start, and we built in several controls to ensure that the delivery of Aβ would be at very low concentrations and not would damage the brain (heart injections).

However, the reviewer raises a critical point about the characterization of endogenous Aβ in zebrafish. To address the point experimentally, we tested three different antibodies to detect endogenous levels of Aβ in zebrafish. We were able to detect very faint bands with the commonly used 4G8 -Aβ antibody (4G8 maps to residues 18–23, conserved between zebrafish Appa and human amyloid beta). However, the sensitivity was not enough to reliably detect endogenous Aβ in either larval or adult zebrafish (Author response image 1). We then ordered a high sensitivity immunoassay kit (V-PLEX Plus Aβ42 Peptide (4G8), which has a dynamic range of 0.516-1271 pg/mL. However, we did not receive it and could not continue our experiments due to the Covid-19 outbreak.

This outcome is consistent with the zebrafish literature. There are many papers using zebrafish for AD research; however, there are no reports of detection of endogenous amyloid beta in larval zebrafish. One paper claims to measure endogenous Aβ levels in larval zebrafish using Westerns (Nery et al., 2017), but bands under 20 kDa are not labelled with a protein ladder making it unclear whether these really correspond to the Aβ band at 4 kDa (or dimers and trimers at ~8 and ~12 kDa or some other species). Wilson and Lardelli developed an assay to measure y-secretase activity in zebrafish but used an artificially modified form of App instead of the endogenous zebrafish App (Wilson and Lardelli, 2013). One study of brain-wide proteomic changes in *bace-1* mutant vs. wild type zebrafish found that zebrafish Appa and Appb proteins accumulate to higher levels in the *bace-1* mutants, suggesting that App processing in the zebrafish is conserved (Hogl et al., 2013). Thus, we believe that detection of zebrafish Aβ is not due to a difference of App processing between species but rather due to the lack of sensitivity of the antibodies.

Although a precise quantification of Aβ in the zebrafish brain remains elusive, the Hogl et al., 2013, proteomics analysis does allow for an estimate of zebrafish Appa and Appb protein levels, as well as the amount they are processed by Bace-1. Benchmarking their reported intensity‐Based Absolute Quantification (iBAQ), which scales proportionally to Molar concentration, for Appa and Appb against one of the most abundant fish brain proteins, Ependymin (estimated at 16 mM in fish ECF), allows for a rough estimate of protein concentration. We estimate from this that Appa is present at 1.7µM and Appb at 3.2µM. Since *bace-1* mutants accumulate 2-fold more Appa and 3-fold more Appb, this suggests that 50-66% of all zebrafish App is processed by *bace-1*. Thus, our estimate of 30-300 pM for the oligomer injections is 4-5 orders of magnitude lower than the concentrations of App, consistent with our other estimates. We recognize that this is only an indirect estimate and does not replace the need for measuring endogenous levels of Aβ and its oligomers at various sites in the zebrafish brain. We do believe it reinforces our contention that the concentrations we have injected are very low and reasonable, especially when compared to other typical in vivo and in vitro studies.

I think the findings are significant and important on their own without having to make this claim which in this case is highly speculative. I would suggest either addressing experimentally or rewording and de-emphasizing this point in the text to make clear the speculative possibilities. In any case, these shortcomings should be more forthrightly noted.

We agree that our findings are important without having to make a strong claim about endogenous processes. We have therefore followed the advice of reviewer 1 and modified our text to tone down the claim that injection of Aβ would reveal endogenous functions of Aβ. Specifically, we changed the following in the text:

The Summary now reads: “Our data indicate that Aβ can trigger a bi-directional sleep/wake switch”

The Introduction now reads: “the various biological effects of Aβ – in health or disease – remain obscure.”

The Introduction now reads: “Aβ may directly modulate sleep-regulatory pathways”

The Introduction now reads: “Isolating the specific biological effects of Aβ has been experimentally difficult.”

The Discussion now reads: “Although the exogenous application of Aβ oligomers in our experiments limit the conclusions we can draw about endogenous functions of Aβ, which in vivo may present with different structure, local concentrations, and kinetics, our bi-directional Aβ modulation of sleep and wakefulness (Figure 6G) predicts that alterations to the relative concentrations of different Aβ oligomeric forms during healthy aging and AD disease progression may have opposing consequences on sleep and wake behavior.”

Finally, we have added the following sentence to the Discussion: “Our model investigates alterations in sleep/wake behavior due to acute changes in exogenously delivered Aβ levels. Thus, it is possible that the sleep/wake effects observed in our study may be different than those observed when Aβ fluctuates over 24hr or when it is chronically accumulating as in AD.”

Reviewer 2:The use of zebrafish to investigate the role of beta amyloid polymers on sleep/wake regulation is potentially interesting as AD patients suffer from insomnia. Here Ozcan and colleagues inject oligomers synthesized in vitro into the fish neonate hearts and fish motion was then recorded and used as a proxy for sleep and wake states. The authors found a correlation between the polymer length and the impact on fish motor and brain activity.

We thank the reviewer for finding our work potentially interesting. For clarity, we would like to highlight some points about measuring sleep and wake states in zebrafish. While fish motion is indeed recorded as a proxy for sleep and wake states, it is well established that inactive bouts in larval fish lasting longer than 1 minute have all of the behavioral criteria for being bona fide sleep, including circadian and homeostatic regulation, characteristic sleep postures, and changes in arousal thresholds in response to stimuli across multiple modalities (Barlow and Rihel, 2017; Prober et al., 2006; Rihel et al., 2010; Zhdanova et al., 2001). Importantly, these bouts are regulated by pharmacology, genes, and neural circuits that are conserved from human to fish (Berridge et al., 2012; Carter et al., 2012; Prober et al., 2006; Zhdanova et al., 2001). We would also like to stress that waking activity (during active bouts, how active is a larva—e.g. taking a stroll or running sprints) is experimentally separable from sleep duration (e.g. see (Rihel et al., 2010).

While the findings are potentially interesting, several points are unclear if concerning to the reviewer:1) First, all the experiments and interpretations rely on overexpression of Aβ polymers, there is no description or investigation in this study of the normal baseline of Aβ accumulation in this species. One would expect to see such data in Figure 1 and Figure 1—figure supplement 1 for example. Is there in fish a night vs. day, sleep vs. night rhythm of Aβ accumulation/expression?

Please see the response to reviewer 1 about the challenges we faced to quantify the level of Aβ in zebrafish. However, even if we were successful in detecting Aβ, it would have been extremely difficult to detect a sleep/wake rhythm of Aβ in zebrafish. To our knowledge, a daily rhythm of Aβ has only been detected in measurement of the ISF/CSF (Holth et al., 2019; Kang et al., 2009; Roh et al., 2012), but this rhythm is not detected in whole tissue homogenates in mammals (e.g.(Kang et al., 2009)). Since the total CSF in one zebrafish larval brain is only ~6.5 nl (Fame et al., 2016), and typical experiments can only remove 1-2 nl of fluid per animal, it was not feasible to extract CSF from 900-5000 larvae to obtain ~5 μl of CSF for a single timepoint analysis, let alone the numbers that would be needed across 24hr.

2) The fish undergo anesthesia and heart perforation and are recorded a few hours later. How do handling, surgical stress, and confounds of prior anesthesia can be eliminated from "sleep-wake" data interpretation?

To control for the effects of both experimental handling artefacts and exogenous protein confounds, we have included in every experiment a reverse Aβ peptide control that was treated prior to injection the same as the Aβ oligomers, then injected into sibling larvae at the same time as the experimental oligomers, under the same handling and surgical stress, and tracked the reverse peptide injected and oligomer injected larvae side by side in the same boxes. To be clear, in every figure, we display a matching set of reverse peptide controls for comparison. Thus, while we cannot rule out synergistic effects between the Aβ oligomers and stress/anesthesia, the effects we describe must be due to the Aβ oligomer type, because this is the only variable that is different from the controls.

Nevertheless, we wanted to know how our handling was affecting the initial behavior. In the original manuscript (Figure 1—figure supplement 1J), we demonstrated that there is a handling effect that lasts for several hours after injection, as seen by comparing PBS injected larvae to larvae placed in the boxes on a prior day. However, we found that our experimental results are unchanged if we omit or include these first 5 hours in our analysis (Figure 1—figure supplement 1K).

To further dissect the effects of anaesthesia, mounting/unmounting, and injecting into the heart, we have added additional experiments (anaesthesia only; mounting only; injection only) to the figure (Figure 1—figure supplement 1H-I). Since each of these produce a similar subsequent behavioral effect that is statistically indistinguishable from each other, we conclude the effects on behavior is due to initial placement of larvae into tracking systems at the beginning of the video-tracking and not the heart-injection, anaesthesia, or mounting.

3) It is hard for the reader to distinguish a specific effect on sleep/wake. Increased or decreased motion could be due to toxicity or specific stimulation of neuronal circuits due to non-physiological presence of exogenous oligomers. The authors try to tackle this issue with c-fos and ERK staining, but Figure 2 shows at least 6 different staining patterns, none of them compared to a sleep/wake baseline of staining. It is quite worrisome to see such a broad over expression of c-fos throughout the brain when Aβ is accumulated. Are the fish having a seizure?Toxicity could lead to reduced motion and even if it's reversible it can still be transient toxicity until oligomers are washed out. Hyperactivity could be due to aspecific overstimulation of neurons as illustrated by c-fos and ERK staining.

We thank the reviewer for raising important points. We now show by 4 new experiments that sleep/wake states induced by Aβ are indeed specific, acutely reversible (e.g. not paralysis/coma) and engage endogenous sleep-wake active neurons and peptides.

A) The reviewer points out that the *c-fos* patterns induced by various Aβ oligomers were not compared to baseline staining during sleep/wake states, making their interpretation more difficult. We now provide *c-fos* staining for larvae just after light ON (ZT1, or 10am, when they are maximally awake), ZT13, when they are awake an hour before lights OFF, and in the middle of the night (ZT 19, or 4am), when larvae are mostly asleep. This data demonstrates that the Aβ^short^ injected fish brain activity pattern is highly similar to the day (baseline awake) samples collected at ZT1 (Figure 2A, C). Although *c-fos* is upregulated in many regions, including the posterior hypothalamus and the dorsal and ventral telencephalon, these areas are also upregulated in the awake larvae at ZT1. We now point to 9 discrete populations that are *c-fos* positive in both sets. This is different to the massive, non-specific upregulation of *c-fos* seen in seizures (Baraban et al., 2005; Reichert et al., 2019). Similarly, *c-fos* expression following Aβ^long^ injections, which was globally dampened relative to Aβ^rev^, is consistently low, similar to the low level of *c-fos* observed in ZT19 sleeping brains.

B) The reviewer asks whether Aβ injected fish have seizures. Seizures in zebrafish can be induced with the epileptogenic drug, PTZ, and are readily detected in larval zebrafish from behavioral analysis (Baraban et al., 2005). WT fish display characteristic “burst-and-glide” swimming movements that are characterized by a short single forward or turning movement followed by a short pause. This bout structure is not detected in animals having seizures, which have uncoordinated bout structures. We now provide tracking data and representative videos for Aβ^rev^, Aβ^short^ and Aβ^long^ injected fish, which all have similar bout structures to uninjected wild type larvae (Figure 1—figure supplement 3D, and Figure 1—video 1). They do not display bouts that resemble seizures (compare to the PTZ treated larvae in Figure 1—figure supplement 3D). Following the analysis of Reichert et al., 2019, we quantified seizure behavior as the presence of high frequency bouts (HFB), which are present in 10 mM PTZ exposed larvae but not untreated larvae, and found that HFBs are not detected in Aβ injected fish, as in uninjected controls (Figure 1—figure supplement 3E).

C) The reviewer suggests that reduced motion seen after injecting Aβ long could reflect toxicity. To test this, we asked whether the inactive states of these larvae were acutely reversible to the same degree as wild type larvae. Larvae in natural sleep states can be aroused into wakefulness by strong salient stimuli, such as a sudden switch in lighting (Prober et al., 2006). Conversely, animals that are paralyzed, in a coma, or displaying sickness behavior (e.g. various toxic effects) do not easily reverse into waking states. We tested this by using 10 min-dark pulses which induces a characteristic response in WT larvae-- rapidly heightened activity when the lights turn off and reversal to baseline activity when the lights turn on. We observe that Aβ^short^ and Aβ^long^ injected fish show a response indistinguishable from WT siblings (Figure 1—figure supplement 3C), indicating that they have not lost their ability to respond to salient stimuli as might be expected from a state of generalized toxicity.

D) If the behavioral states induced by Aβ are indeed real sleep/wake states, oligomers should engage known sleep/wake regulatory pathways. Galanin-expressing neurons are active and *galanin* expression is increased during sleep in larval zebrafish (Reichert et al., 2019). Thus, we tested whether Aβ injections alter the number of strongly *galanin* expressing neurons in the hypothalamus, which correlates with sleep state (Reichert et al., 2019). We did in situ hybridization (ISH) for *galanin* 4-6 hours post-injection. All staining steps for different groups was done in the same tube to eliminate variations in ISH procedure, and subsequent neuron counts were done blindly. Consistent with Aβ^long^ inducing sleep and Aβ^short^ inducing wakefulness, we observed a slight decrease (-6%) in hypothalamic *galanin* cell number in Aβ^short^ injected fish and a slight increase (+12%) in Aβ^long^ injected animals compared to Aβ^rev^ injected fish (Figure 2F-G). From blinded counts, *galanin* cell numbers between Aβ^short^ and Aβ^long^ injected fish were significantly different from each other (p<0.01, Kruskal-Wallis). We have included represented images and analysis of these *galanin* changes in Figure 2F-G.

4) Injections in mutant backgrounds indeed show some specificity in binding/interaction but still it does not demonstrate that the impact is on wake or sleep regulation per se. Again only motion or broad brain staining (at one time point…) are shown. An alternative interpretation is that adrb2a, pgrmc, prp1 can indeed bind Aβ but relay the toxic or aspecific impact of oligomers over expression in a brain that normally does not accumulate such molecules.

Please see our points addressing comment 3 that detail our attempts to demonstrate the impact on sleep/wake regulation per se, especially the induction of wild type sleep/wake *c-fos* patterns and effects on sleep circuits like *galanin.* The *c-fos* patterns induced by oligomers is similar to changes in WT *c-fos* patterns over 24 hours, which is consistent with an interpretation that oligomers induce normal brain state changes. Additionally, Aβ^long^ engages known sleep-regulatory pathways such as *galanin*, and *galanin* expression is reduced in Aβ^short^ injected animals, which is also consistent with these changes being sleep/wake states. Together, these observations suggest that the induced states are not general toxicity but rather acute signalling events.

However, we take the point that it is difficult to completely rule out “toxicity” – for example even engagement of sleep circuits could be due to these circuits being the site of “toxic” effects. Even so, given the changes in Aβ composition in Alzheimer’s Disease patients, the demonstration of acute effects on sleep-regulating circuits should still be of wide interest.

We have added some more caveats to our interpretation to the Discussion section. Specifically, we added: “Although the exogenous application of Aβ oligomers in our experiments limit the conclusions we can draw about endogenous functions of Aβ, which in vivo may present with different structure, local concentrations, and kinetics, our bi-directional Aβ modulation of sleep and wakefulness (Figure 6G) predicts that alterations to the relative concentrations of different Aβ oligomeric forms during healthy aging and AD disease progression may have opposing consequences on sleep and wake behavior.” and “Our model investigates alterations in sleep/wake behavior due to acute changes in exogenously delivered Aβ levels. Thus, it is possible that the sleep/wake effects observed in our study may be different than those observed when Aβ fluctuates over 24hr or when it is chronically present as in AD”.

This study has the potential to be extremely interesting but many controls and demonstration of endogenous Aβ role on sleep-wake cycle are needed.Reviewer 3:[…] 1) While the experiments are well performed and the data intrinsically consistent, the applicability to mammals (and humans) is a consideration. Infusion of Aβ into the heart of larvae is a highly artificial system, and events that occur during sudden changes in Aβ levels may be different that those observed when Aβ is chronically present (as in AD). For example, infusion of Aβ peptide into the brain of mice or rats can induce acute, local neurodegeneration that is not observed in APP transgenic mice with chronically elevated Aβ levels. This is a fundamental shortcoming of the model, and there is little that can be done to address it, but it should be perhaps mentioned in the Discussion.

We would like to point out that we cannot detect any increase in neurodegeneration in our study (Figure 1—figure supplement 2), unlike the examples mentioned from injections in the brain of mice and rats. However, we agree our method has shortcomings relative to endogenous manipulations. We have made many changes to the text to make this shortcoming clearer. Please see responses to reviewer 1 and 2 for detailed examples of these changes.

2) The implications of this bidirectional effect of short and long oligomers for sleep phenotypes in AD are also a bit unclear, as oligomers of all sizes are likely present in AD brain (though perhaps in different ratios as the disease progresses). It would be helpful to determine which pathway is dominant when both short and long oligomers are infused together, perhaps in different ratios. This is the only experiment I would request.

Thank you for the suggestion for this experiment, which we believe also provides insight into whether the Aβ^short^ and Aβ^long^ are signalling onto the same or distinct neuronal circuit pathways or signalling cascades. To address which molecular pathway dominates if both short and long oligomers are present, we injected Aβ^rev^, Aβ^short^ and Aβ^long^ alone or the short and long oligomers mixed in a one to one ratio (Aβ^mix^). Confirming our previous results, the short and long oligomers gave opposite sleep phenotypes (p<0.01, Kruskall-Wallis) in 4 new independent experiments. The sleep-wake phenotype of the Aβ^mix^ lies between the Aβ^short^ and Aβ^long^ injected animals (Figure 6F), consistent with an additive effect of the two oligomers. This suggests that the long and short oligomers affect independent neuronal pathways/signalling cascades in addition to acting through distinct receptors (Adrb2a/Pgrmc1 vs. Prp). Future experiments will be needed to map the specific circuits involved, for example by testing whether systematic replacement of the receptors into subsets of neuronal circuits on mutant backgrounds can rescue the effects of Aβ oligomers.

We have added the following lines to describe this new data: “Co-injection of both Aβ^short^ and Aβ^long^ resulted in an intermediate phenotype that is indistinguishable from control injections of Aβ^rev^ (Figure 6F), suggesting that the effects of Aβ^short^ and Aβ^long^ are additive and likely act through distinct neuronal circuits or signalling cascades to modulate sleep. Considering this result together with the genetic and pharmacological data, we propose a bi-directional model of Aβ sleep regulation in which Aβ^short^ and Aβ^long^ act through distinct receptors and neuronal pathways to independently modulate behavioral state (Figure 6G).”

References:

Berridge, C.W., Schmeichel, B.E., Espana, R.A., 2012. Noradrenergic modulation of wakefulness/arousal. Sleep Med Rev 16, 187-197.

Carter, M.E., Brill, J., Bonnavion, P., Huguenard, J.R., Huerta, R., de Lecea, L., 2012. Mechanism for Hypocretin-mediated sleep-to-wake transitions. Proc Natl Acad Sci U S A 109, E2635-2644.

Fame, R.M., Chang, J.T., Hong, A., Aponte-Santiago, N.A., Sive, H., 2016. Directional cerebrospinal fluid movement between brain ventricles in larval zebrafish. Fluids Barriers CNS 13, 11.

Hogl, S., van Bebber, F., Dislich, B., Kuhn, P.H., Haass, C., Schmid, B., Lichtenthaler, S.F., 2013. Label-free quantitative analysis of the membrane proteome of Bace1 protease knock-out zebrafish brains. Proteomics 13, 1519-1527.

Holth, J.K., Fritschi, S.K., Wang, C., Pedersen, N.P., Cirrito, J.R., Mahan, T.E., Finn, M.B., Manis, M., Geerling, J.C., Fuller, P.M., Lucey, B.P., Holtzman, D.M., 2019. The sleep-wake cycle regulates brain interstitial fluid tau in mice and CSF tau in humans. Science 363, 880-884.

Nery, L.R., Silva, N.E., Fonseca, R., Vianna, M.R.M., 2017. Presenilin-1 Targeted Morpholino Induces Cognitive Deficits, Increased Brain Aβ1-42 and Decreased Synaptic Marker PSD-95 in Zebrafish Larvae. Neurochem Res 42, 2959-2967.

Reichert, S., Pavon Arocas, O., Rihel, J., 2019. The Neuropeptide Galanin Is Required for Homeostatic Rebound Sleep following Increased Neuronal Activity. Neuron 104, 370-384.e375.

Rihel, J., Prober, D.A., Schier, A.F., 2010. Monitoring sleep and arousal in zebrafish. Methods Cell Biol 100, 281-294.

Wilson, L., Lardelli, M., 2013. The development of an in vivo gamma-secretase assay using zebrafish embryos. J Alzheimers Dis 36, 521-534.

Zhdanova, I.V., Wang, S.Y., Leclair, O.U., Danilova, N.P., 2001. Melatonin promotes sleep-like state in zebrafish. Brain Res 903, 263-268.